# Bamboo-Based Carbon/Co/CoO Heterojunction Structures Based on a Multi-Layer Periodic Matrix Array Can Be Used for Efficient Electromagnetic Attenuation

**DOI:** 10.3390/ma17215239

**Published:** 2024-10-28

**Authors:** He Han, Hui Chen, Rui Wang, Zhichao Lou

**Affiliations:** 1Jiangsu Co-Innovation Center of Efficient Processing and Utilization of Forest Resources, College of Materials Science and Engineering, Nanjing Forestry University, Nanjing 210037, China; 18805166615@163.com (H.H.); punch.te@icloud.com (R.W.); 2Joint International Research Lab of Lignocellulosic Functional Materials, College of Light Industry and Food Engineering, Nanjing Forestry University, Nanjing 210037, China; huichen@njfu.edu.cn

**Keywords:** bamboo chip, honeycomb-like matrix, heterojunction, multi-interface polarization, microwave absorption

## Abstract

With the popularization of wireless communication, radar, and electronic devices, the hidden harm of electromagnetic radiation is becoming increasingly serious. The design of green biomass carbon-based interface heterojunctions based on lightweight porous materials can effectively protect against electromagnetic radiation hazards. In this work, we constructed an anisotropic heterojunction interface with magnetic and dielectric coupling based on a honeycomb-like periodic matrix multi-layer array repeating unit. The removal of lignin components from bamboo through oxidation enriches the impregnation pores and uniform adsorption sites of the magnetic medium. Further, in situ pyrolysis promotes the formation of a large number of electric dipoles at the interface between the magnetic medium and dielectric coupling inside the periodic cell carbon skeleton, enhancing interface polarization and relaxation. Local carrier traps and uneven electromagnetic density enhance dielectric and hysteresis losses, resulting in excellent impedance matching. Therefore, the obtained bamboo-based carbon multiphase composite absorbent has satisfactory electromagnetic loss characteristics. At a thickness of 1.55 mm, the effective absorption bandwidth reaches 5.1 GHz, and the minimum reflection loss (RL) value reaches −54.7 dB. In addition, the far-field radar simulation results show that the sample has an excellent RCS (radar cross-section) reduction of 33.3 dB·m^2^. This work provides new directions for the diversified development of green biomass and the optimization of the design of magnetic and dielectric coupling in periodic array structures.

## 1. Introduction

With the continuous development of modern electrical technology and wireless communication, the density and intensity of electromagnetic radiation in daily life continue to increase, leading to serious invisible electromagnetic pollution problems [1,2,3]. Long-term exposure to high-intensity electromagnetic radiation may pose a threat to human health, electronic device performance, and information security. For example, electromagnetic radiation may cause overheating of human tissues and cellular dysfunction, as well as interference with the normal operation of sensitive electronic devices and even damage to precision equipment and information loss [4,5,6]. As a key means in the field of electromagnetic protection, electromagnetic absorbers can convert complex electromagnetic radiation in space into other forms of energy for dissipation (such as thermal energy) through mechanisms such as dielectric polarization loss, hysteresis loss, conductive eddy current loss, interface scattering loss, and absorption impedance matching, thereby reducing secondary reflection and transmission pollution of electromagnetic waves and reducing and eliminating electromagnetic radiation interference and pollution hazards [7,8,9]. Therefore, how to effectively absorb and shield electromagnetic waves and develop electromagnetic wave absorbing materials with characteristics such as broadband, lightweight, multiple loss mechanisms, and environmental friendliness has become an important issue that urgently needs to be solved [10,11].

In recent years, many absorbent materials have been developed to address the increasingly severe electromagnetic radiation in space. These include two-dimensional materials such as graphene and two-dimensional transition metal carbides (MXene) that are lightweight, highly dielectric, and have strong weather resistance [12,13]; foam metal (foam iron, foam nickel), foam ceramic, foam carbon, and other porous structural materials with large specific surface areas and multiple reflection paths [14,15]; and magnetic metal materials that exhibit significant magnetic loss effects in the low-frequency range, including iron, cobalt, nickel, and their alloys or oxides. They also include the development and utilization of flexible and biodegradable green polymers (Polylactic Acid (PLA), Poly(butylene succinate-co-adipate) (PBSA), etc.)/carbon-based composites (carbon nanotubes, graphene, etc.) [16,17,18], and lightweight, easy to process, and flexible polymer-based composite materials such as polyaniline (PANI) and polypyrrole (PPy) [19,20]. Although the above materials have a certain absorption effect, they still have significant limitations. Conventional carbon-based materials have poor low-frequency absorption performance; a single absorption mechanism is insufficient in broadband absorption performance, and high conductivity leads to increased reflection. The density of foam metal and magnetic metal particles is large and easy to oxidize; the preparation process is complex, energy consumption is large, and the uncontrollable structure reduces the electromagnetic adjustment ability [21,22]. Conductive polymers are prone to aggregation, and poor dispersibility reduces reflection losses. Considering the efficient multiple loss mechanism of electromagnetic radiation, it is necessary to overcome the shortcomings of traditional absorbent materials in order to further integrate the compositional characteristics of materials and optimize the structure of unit cells [23,24,25].

It is worth noting that magnetic nanoparticles have a higher proportion of surface magnetic atoms due to size and surface effects, resulting in stronger magnetic loss ability. At the same time, the magnetic domain size of nanoparticles is smaller than or close to the single domain size, and under the action of electromagnetic waves, magnetic domain resonance is more significant, which can enhance the absorption of electromagnetic waves. In addition, when lightweight magnetic nanoparticles are combined with other lightweight substrates such as carbon-based materials, the magnetic loss mechanisms such as domain wall movement, hysteresis loss, and the eddy current effect in the magnetic material can work together with various loss mechanisms such as dielectric loss and interface polarization to form a composite absorption mechanism and expand the absorption frequency range of materials. At the same time, it makes the impedance of electromagnetic waves between the material surface and free space more matched, enhancing absorption efficiency. However, there are still significant challenges in constructing lightweight composite absorbers with green, low-cost, and efficient coupling of magnetic and dielectric materials for multi-heterojunction interface polarization.

Biomass resources in nature, such as bamboo, wood, seaweed, and agricultural waste, are abundant and renewable. Compared to traditional metal and polymer-based materials, they have natural sustainability advantages, such as being low-carbon, environmentally friendly, biodegradable, and having low processing costs [26,27,28]. Among them, bamboo has a shorter growth cycle and more abundant resources. Compared to other carbon-based materials, bamboo is composed of regularly arranged thin-walled cells and vascular bundles, forming a highly ordered honeycomb-like multi-layer matrix array microstructure. This structure endows bamboo-based carbon materials with good mechanical strength and low density [29,30]. The honeycomb-like microstructure itself can provide materials with a natural multi-level pore structure, enriching the multiple reflection and scattering effects of electromagnetic waves. Bamboo has a higher specific surface area and flexibility than does other biomass [31,32,33]. The abundant natural polymers (cellulose, hemicellulose) inside contain various functional chemical groups (hydroxyl, carboxyl, methoxy, etc.), which provide multiple reaction sites for chemical modification, composite material preparation, and functional applications of bamboo. Also, after carbonization treatment, the structure can maintain its original strength and toughness well, forming a stable bamboo-based carbon material [34,35]. Therefore, low-cost biomass-based lightweight porous electromagnetic absorbers with efficient coupling of magnetic and dielectric media can be obtained through a simple preparation process. Then, in this process, the reasonable and efficient loading of magnetic media (such as ferrite, cobalt, nickel, and other magnetic metals or their oxides) is crucial [1,2]. Uneven distribution or agglomeration of magnetic nanoparticles in porous carbon matrices can lead to weakened interfacial polarization effects, deteriorated impedance matching, reduced multiple scattering and magnetic loss effects, and narrowed effective absorption frequency bands, thereby affecting the material’s absorption efficiency and broadband absorption capacity [36,37]. Reunion will also reduce the lightweight advantage of materials. On the contrary, the introduction of reasonable magnetic particles can generate more anisotropic heterojunction interface structures within bamboo-based carbon. In addition, the introduction of magnetic media can further alleviate the thermal decomposition or expansion of bamboo-based carbon materials at high temperatures, improve the thermal expansion coefficient of the material, and enhance structural stability. Therefore, it is crucial to achieve effective integration of magnetic media inside bamboo [8,38].

Here, we removed lignin components that hinder impregnation diffusion inside bamboo through oxidation modification while selectively adjusting the abundant functional chemical groups on the surface of the cellulose and hemicellulose. A series of bamboo-based carbon/Co/CoO multi-interface heterojunction electromagnetic absorbers with multi-layer periodic matrix arrays were obtained by introducing cobalt-containing solutions of different concentrations into bamboo through vacuum impregnation and in situ pyrolysis. The change in concentration results in different magnetic medium ratios, defect concentrations, and dielectric properties. Further research has shown that magnetic media and dielectric coupling interfaces can cause charge redistribution and form electric dipoles under the action of electromagnetic fields, further enhancing interface polarization and relaxation. The formation of local carrier traps and uneven electromagnetic energy density at heterojunction interfaces enhances local dielectric losses and eddy current effects. By optimizing the heterojunction structure, excellent magnetic loss and dielectric coupling effects are constructed to obtain suitable impedance-matching characteristics. Efficient electromagnetic wave absorption can be achieved over a wider frequency range. This work provides new directions for the diversified development of green biomass and optimization of the design of magnetic and dielectric coupling in periodic array structures.

## 2. Materials and Methods

### 2.1. Materials

The bamboo slices required for the experiment were selected from Nanping City, Fujian Province, China. CO (NO_3_)_2_·6H_2_O and ethanol (EtOH) were purchased from Shanghai Aladdin Biochemical Technology Co., Ltd., Shanghai, China. NaClO_2_ (w%, 80%) and acetic acid (w%, 99.5%) were purchased from Shanghai McLean Biochemical Co., Ltd., Shanghai, China.

### 2.2. Removal of Lignin from Bamboo Slices

Weigh 15 g of NaClO_2_ and 250 mL of deionized water to prepare a homogeneous solution and adjust the pH of the NaClO_2_ aqueous solution to 4.5–4.7 by adding acetic acid. Place the cut 4 cm × 4 cm × 1 mm bamboo slices in the above solution under the reaction conditions of heating in an oil bath at 80 °C for 3 h. After the reaction is complete, wash the bamboo slices alternately with ionized water and ethanol until the residue is completely removed. Finally, dry in a 40 °C vacuum oven for 12 h. Name it BD and take it out for later use.

### 2.3. Preparation of Bamboo-Based Carbon/Co/CoO Composite Absorbing Material

Prepare 0.05 M, 0.1 M, 0.15 M, and 0.2 M Co (NO_3_)_2_·6H_2_O aqueous solutions in beakers, and place an appropriate amount of lignin-removed bamboo slices (BD) in the above solutions for vacuum immersion for 4 h. Afterward, take out the bamboo slices and dry them at 50 °C for 8 h. Subsequently, samples immersed in different concentrations of Co^2+^ were placed in a tube furnace and heated to 1000 °C at a heating rate of 5 °C/min under N_2_ protection for 2 h; the samples were then cooled to room temperature to obtain a series of bamboo-based carbon/Co/CoO composite absorbing materials (BCs), which were named BC-1 BC-2, BC-3, BC-4.

### 2.4. Evaluation of Electromagnetic Absorption Performance

Calculate the minimum reflection loss RL value of BCs by transmission line theory:(1)$$Z_{in}=Z_{0}\sqrt{\mu_{r}/\varepsilon_{r}}tanh[j(2\pi fd/c)\sqrt{\mu_{r}\varepsilon_{r}}]$$
(2)$$RL\left(dB\right)=20\log\left|(Z_{in}-Z_{0}\right)/(Z_{in}+Z_{0})|$$

Among them, the complex permittivity ($\varepsilon_{r}=\varepsilon^{\prime }+j\varepsilon^{\prime \prime }$), where *j* represents the symbol of the imaginary part, $\varepsilon^{\prime }$ and $\varepsilon^{\prime \prime }$ represent the real part of the permittivity and the imaginary part of the permittivity and the permeability ($\mu_{r}=\mu^{\prime }+j\mu^{\prime \prime }$), $\mu^{\prime }$ and $\mu^{\prime \prime }$ represents the real part of the permeability and the permeability of the imaginary part. $Z_{in}$ and $Z_{0}$ are the free space impedance values of the electromagnetic wave absorbing material and the input impedance, *f* represents the frequency of material testing, *c* represents the speed of light, and *d* represents the matching thickness.

### 2.5. Characterization

Use environmental scanning electron microscopy (SEM, JSM-7800F, Tokyo, Japan) to characterize the microscopic morphology and structural characteristics of BC samples. Analyze the EDS elemental composition of the sample using an Oxford energy spectrometer (XMX1236, Tokyo, Japan). The internal chemical composition and elemental properties of the BC samples were determined by X-ray photoelectron spectroscopy (XPS, Thermo Scientific, ESCALAB 250Xi, Palo Alto, CA, USA) and X-ray diffraction (XRD, Bruker D8 ADVANCE, Sunnyvale, CA, USA). Test the microstructure and lattice characteristics of the sample through transmission electron microscopy. The magnetic properties of BC samples were tested using a vibrating sample magnetometer (VSM, PPMS-9T, Wilmington, DE, USA). Raman spectroscopy was performed using a Renishaw sample collection system equipped with a 532.0 mm 50 mW DPSS laser in a Via Plus micro-Raman spectrometer, Wilmington, DE, USA. The electromagnetic wave absorption performance and corresponding electromagnetic parameters of the material were tested using an Agilent PNA N5224A, Palo Alto, CA, USA. vector network analyzer, in which the powder sample was mixed with paraffin in a ratio of 1:4 and pressed into a specification ring sample. The frequency range for testing is 2–18 GHz, and the testing method is the coaxial line method (Figure 1).

## 3. Results

In this work, we preprocessed bamboo slices using a delignification process, as shown in Figure 2a–d, which are SEM images of the original bamboo slices and those after lignin removal. Untreated thin-walled cells are tightly arranged, and the cell surface and intercellular spaces are filled with substances such as starch, lipids, wax, tannins, and sugars. Meanwhile, lignin mainly exists between the cellulose and hemicellulose, filling the gaps in the cell wall. After oxidation modification treatment, the basic morphology of thin-walled cells did not change; the cell wall became thinner, and the intercellular space increased. Some cells showed slight concave–convex deformation and a small amount of interlayer displacement (Figure 2d). This indicates that after removing the lignin from bamboo, a smooth cobalt salt impregnation channel is formed inside. This further promotes the absorption and loading of cobalt ions. Furthermore, through in situ pyrolysis treatment of bamboo slices impregnated with cobalt salts, we found that a large number of uniformly dispersed nanoparticles grew on the surface and inside of thin-walled cells in the BC-3 bamboo radial section (Figure 2g), indicating that cobalt salts are completely infiltrated inside bamboo and grow uniformly under the interaction force of internal functional groups. At the same time, the carbon skeleton structure composed of thin-walled cells did not show significant changes (Figure 2f). The corresponding EDS elemental analysis indicates that the particles mainly contain the cobalt element. By observing the cross-sectional morphology of the BC samples, we found that the cells still maintained a neatly arranged honeycomb-like hexagonal matrix array structure (Figure 2n) and were loaded with a large number of cobalt-containing nanoparticles on the surface. The corresponding element distribution is consistent with the results of the bamboo slice diameter section. The weak oxygen element is mainly attributed to the thermal decomposition reaction of some substances in bamboo at high temperatures. It is worth noting that by changing the solubility of different impregnating solutions, the number of magnetic nanoparticles loaded on the cell surface gradually increases, and partial particle aggregation phenomena gradually occur (Appendix A). In addition, the change in concentration did not cause any changes in the morphological arrangement of the thin-walled cells inside the bamboo and still maintained a relatively neat multi-layer matrix structure. The above results indicate that the obtained cobalt-containing nanoparticles are formed by the interaction between Co(NO_3_)_2_·6H_2_O and the bamboo-based carbon matrix under high-temperature pyrolysis. In order to further analyze the main components of the particles, as shown in Figure 2k–m, we characterized the TEM lattice structure of the BC-3 sample. The lattice spacing is indexed by two adjacent fingers, corresponding to the (111) plane with a lattice spacing of 0.195 nm and the (200) plane with a lattice spacing of 0.212 nm, respectively, corresponding to the lattice structures of Co and CoO [39]. The selected area electron diffraction (SAED) image shown in Figure 2k shows that the bright spots of the diffraction ring represent the (110), (200), and (220) lattice planes of Co, respectively [40]. The results indicate that the nanoparticles on the surface of thin-walled cells are mainly composed of Co and CoO. By oxidizing bamboo to remove lignin, a large number of voids and microporous structures are formed between cells, further promoting the full immersion of the cobalt salt solution. Directed regulation of the abundant functional chemical groups on the surface of cellulose and hemicellulose promotes the uniform dispersion and loading of magnetic nanoparticles.

To further confirm the compositional information of the nanoparticles, we conducted XRD testing on the material, as shown in Figure 3a–f. The absorption peak corresponds to the (002) plane (g-C, JCPDS 75-1621) of graphite carbon at 2θ = 26.2° [7]. As the immersion concentration increases, the absorption peak gradually weakens. This may be attributed to the large number of magnetic nanoparticles loaded on the surface and interior of the material cell wall, which reduces the signal intensity of graphite carbon. Taking BC-3 and BC-4 as examples, we found absorption peaks at 2θ = 41.78°, 44.37°, 47.31°, 51.59°, and 76.03° in the images, corresponding to the (100), (002), (101), (200), and (220) planes of cobalt, respectively (JCPDS No.15-0806) [41]. At the same time, the peak at 2θ = 42.38° is a representative diffraction peak of the cobalt-oxide (200) plane (JCPDS No. 48-1719) [42]. These results indicate that the BC series samples all generated Co and CoO magnetic nanoparticles. The introduction of magnetic nanoparticles can, to some extent, affect the internal defects of carbon-based composites, so we studied the graphitization degree of BC series samples through Raman spectroscopy. From the figure, it can be seen that the two absorption peaks represent the D band (around 1350 cm^−1^, corresponding to the sp3 atomic vibration of disordered graphite) and the G band (around 1600 cm^−1^, corresponding to the in-plane vibration of sp2 atoms in a 2D hexagonal lattice), respectively. The value of ID/IG can reflect the degree of graphitization of the material. With the continuous increase of immersion concentration, the ratio of D peak to G peak ID/IG gradually increases, and BC-4 has the maximum value, reaching 0.86. This indicates that concentration has a promoting effect on the graphitization degree of composite material systems and can effectively regulate the interface defects inside the materials [43]. This is attributed to the loading of a large number of nanoparticles that disrupted the graphite interface structure. Furthermore, XPS spectroscopy was used to analyze the types of elements and changes in functional groups within the material. From Figure 3c, it can be seen that characteristic peaks corresponding to the three elements C, O, and Co appear in the figure. Specifically, by observing the high-resolution XPS Co2p spectra and their curve-fitting images, we found four deconvolution peaks at 778.6 eV, 794.9 eV, 781.5 eV, and 797.5 eV, corresponding to the binding energy information of Co^0^ and Co^2+^, respectively [44]. Four deconvolution peaks were observed at 784.0 eV, 801.7 eV, 786.8 eV, and 803.2 eV, corresponding to the binding energy absorption peaks of the plasma loss peak and satellite peak, respectively. Furthermore, we conducted high-resolution XPS spectroscopy analysis of C1s and O1s in the C-O complex of BCs. As shown in Figure 3e,f, three deconvolution peaks were observed at 530.1, 532.0, and 533.3 eV, representing the binding energies of Co-O, C-O, and C=O bonds, respectively. The three deconvolution peaks appearing at 284.8, 285.83, and 288.0 eV represent the binding energy absorption peaks of Co-O, C-O, and C=O bonds, respectively [45]. It is worth noting that as the immersion concentration increases, the intensity ratio of the C=O and C-O absorption peaks of the four samples changes, indicating that the loading of magnetic nanoparticles can, to some extent, regulate the defect polarization performance inside bamboo. At the same time, the O atoms in the C=O dipole bond have more negative charges than those in the C-O bond, and the interface will cause a redistribution of charges and form electric dipoles under the action of electromagnetic fields, further enhancing interface polarization and relaxation. The above results indicate that bamboo slices impregnated with cobalt salts have undergone carbonization treatment, resulting in a multiphase bamboo-based carbon/Co/CoO heterostructure with Co and CoO magnetic nanoparticles.

Through in situ pyrolysis treatment, a large number of magnetic nanoparticles are generated inside the cellular carbon skeleton. The formation of a large number of electric dipoles at the interface between internal magnetic media and dielectric coupling enhances interface polarization and relaxation. Local carrier traps and uneven electromagnetic density enhance dielectric and hysteresis losses and promote the absorption of electromagnetic waves. In order to visually describe the electromagnetic wave absorption performance of the samples, we calculated the reflection loss (RL) values of the samples in the frequency range of 2.0 to 18.0 GHz and the matching thickness range of 1.00 to 5.00 mm based on their respective electromagnetic parameters and plotted the relevant images. As shown in Figure 4a–d, we found that the BC-3 sample has the best reflection loss at 17.8 GHz, with an RL value of −54.7 dB. The matching thickness is only 1.35 mm. This indicates that the BC-3 sample has the best electromagnetic wave absorption effect, and the ultra-thin thickness has better commercial application value. At the same time, as the concentration increases, the reflection loss RL value of the BC samples shows a trend of first decreasing and then increasing. This indicates that the concentration of magnetic nanoparticles has a certain regulatory effect on the electromagnetic wave absorption performance of the sample. At the same time, the minimum values of the reflection loss RL of the BC samples are all less than −10 dB, with RL values of −37.3 dB (matching thickness 4.8 mm), −40.8 dB (2.1 mm), −54.7 dB (1.35 mm), and −17.5 dB (3.45 mm), respectively, indicating that the material can attenuate and dissipate 90% of the electromagnetic wave energy incident inside the material. Excellent electromagnetic wave absorbing materials not only need to have matching thickness and excellent reflection loss RL values; it is equally important for them to effectively absorb the bandwidth under thin thickness. As shown in Figure 4e–h, the optimal effective absorption bandwidth of the BC samples are 1.96 GHz (4.8 mm), 3.64 GHz (2.35 mm), 5.1 GHz (1.55 mm), and 3.96 GHz (4.25 mm), respectively. The results show that the BC-3 sample has a satisfactory effective absorption bandwidth, with not only the widest EAB (Effective absorption bandwidth) but also the thinnest corresponding matching thickness. This indicates that the BC-3 sample has an excellent electromagnetic response mechanism, which can meet the attenuation loss requirements of electromagnetic radiation in different application scenarios and frequency bands. The repetitive unit structure of the periodic matrix array inside bamboo can form an electromagnetic bandgap, which prevents electromagnetic waves in certain frequency ranges from passing through, thereby improving the absorption performance. At certain frequencies, under the influence of appropriate periodic unit sizes and electromagnetic parameters of the material, local resonance may occur inside the material, which enhances the absorption of electromagnetic waves to further elucidate the total effective absorption bandwidth of BC samples under multiple matching thicknesses, as shown in Figure 4i–l. The total EAB of sample BC-3 reached 13.3 GHz in the frequency range of 2.0 to 18.0 GHz and the matching thickness range of 1.00 to 5.00 mm, while BC-2 and BC-4 only reached 11.8 GHz and 7.9 GHz. These results indicate that effective absorption of electromagnetic waves from low to high frequencies can be achieved by adjusting the thickness of the sample preparation. Furthermore, the magnitude of the attenuation constant plays an equally important role in the dissipation of electromagnetic energy. Specifically, the attenuation constant can reflect the EM (electromagnetism) energy attenuation ability of the absorber. The higher the attenuation constant, the stronger the ability to convert electromagnetic energy into other forms of energy. Overall, BC-3 samples exhibit the best attenuation constant values within the frequency range of 2.0 to 18.0 GHz. This indicates that it has satisfactory electromagnetic wave absorption and dissipation characteristics (Appendix A).

According to formulas $Z_{in}=Z_{0}\sqrt{\mu_{r}/\varepsilon_{r}}tanh[j(2\pi fd/c)\sqrt{\mu_{r}\varepsilon_{r}}]$ and $RL\left(dB\right)=20\log\left|(Z_{in}-Z_{0}\right)/(Z_{in}+Z_{0})|$, we found that the change in the magnitude of the electromagnetic parameter ($\varepsilon^{\prime },\varepsilon^{\prime \prime },\mu^{\prime }$ and $\mu^{\prime \prime }$) of the material affects the reflection loss RL value and impedance-matching characteristics of the material. Therefore, we elaborated on the electromagnetic attenuation mechanism of the material by comparing the peak values and variation patterns of various electromagnetic parameters of BCs. As shown in Figure 5a–d, the electromagnetic parameter ($\varepsilon^{\prime },\varepsilon^{\prime \prime },\mu^{\prime }and\mu^{\prime \prime }$) exhibits varying degrees of variation in the frequency range of 2.0–18.0 GHz. Specifically, the BC-3 sample exhibits the best real part (8.94–16.25) and imaginary part values (1.67–7.47) of dielectric constant across the entire frequency range. The maximum values reached 16.25 and 7.47, respectively. Generally speaking, the complex dielectric constant of a material can be represented by formula $\varepsilon_{r}=\varepsilon^{\prime }+j\varepsilon^{\prime \prime }$, where $\varepsilon^{\prime }$ represents the storage effect of absorbing materials on external electric field charges, which to some extent affects the interface polarization of the material; $\varepsilon^{\prime \prime }$ represents the energy loss caused by the disruption and recombination of the electric dipole moment inside the material under the promotion of an alternating electric field. Therefore, compared to other samples, the rich heterojunction structure inside BC-3 leads to the accumulation and separation of charges at the interface, resulting in strong polarization effects and wideband relaxation phenomena. The synergistic effect of interface polarization and relaxation phenomenon enhances the dielectric loss of the material. Make the material have excellent absorption performance for electromagnetic waves in a wide frequency range. It is worth noting that the impregnation solution concentration of BC-4 is the highest, but the real part (2.51) and imaginary part (0.03) of the complex dielectric constant are the smallest. This may be attributed to the generation of a large number of cobalt-based magnetic nanoparticles inside the material under high-temperature pyrolysis, which interact with the carbon matrix and produce abundant interface defects, affecting the conductivity of the carbon matrix. High concentrations of Co and CoO may cause particles to aggregate in the material, forming large-scale conductive networks or clusters. This cluster structure may suppress interface polarization and dipole polarization, reduce the energy storage and loss capacity of the material, and lead to a decrease in the dielectric constant. In addition, the BC sample generates multiple fluctuation peaks throughout the frequency range, and the resonance peak amplitude of BC-3 is the largest. Under the action of alternating electromagnetic fields, BC-3 exhibits richer multiple relaxation polarizations and has the best electromagnetic losses. Meanwhile, the dielectric loss tangent (*tanδε*) parameter of the material also shows a similar trend of change. The dielectric loss tangent (*tanδε*) of BC-3 is the highest, reaching 0.74 (Appendix A). The introduction of magnetic media significantly improves the electromagnetic loss effect of materials. As the immersion concentration increases, we can see that the real part (0.81–1.39) and imaginary part (0.12–0.95) of the magnetic permeability of BC-4 material have maximum values, reaching 1.39 and 0.95, respectively. This may be attributed to the significant enhancement of the magnetic properties of BC-4 material by the abundant magnetic nanoparticles (Appendix A). The imaginary part of the BC-3 sample shows a negative value, which is due to the effect of magnetic energy radiation. The electromagnetic field at high frequencies is greatly enhanced, which interferes with the magnetic losses inside the material. $\mu^{\prime \prime }$ fluctuates and decreases to a negative value. Meanwhile, the image of the magnetic loss tangent (*tanδμ*) shows that the optimal value for BC-4 is 0.77 (Appendix A). These results indicate that the magnetic loss of BC-4 samples plays an important role in the entire loss component system. The excellent electromagnetic wave absorption performance of BC-3 is mainly attributed to its optimal dielectric loss characteristics. With the increase of Co and CoO concentrations, the coupling effect between magnetic particles is enhanced, especially the eddy current loss and hysteresis loss, which are more significant at high concentrations. However, excessive concentration may cause conductivity loss and particle aggregation effects, which suppress polarization and lead to a decrease in the dielectric constant; on the contrary, it suppresses the absorbing effect of the material. From image $C_{0}$ ($C_{0}=\mu^{\prime \prime }{\left(\mu^{\prime }\right)}^{-2}f^{-1}$) in Appendix A, it can be observed that the BC sample exhibits significant fluctuations across the entire frequency range, indicating that magnetic losses mainly come from exchange resonance and natural resonance and are not related to eddy current losses.

By conducting a rational optimization design of multiple components, the electromagnetic parameters of different samples were adjusted, further affecting the impedance-matching characteristics of the material. This is crucial for electromagnetic wave-absorbing materials. The closer the value of $\left|Z_{in}/Z_{0}\right|$ is to one, the easier it is for the material to absorb electromagnetic radiation from space. When the value of $\left|Z_{in}/Z_{0}\right|$ is less than or greater than one, the reflection effect of the material on spatial electromagnetic waves increases, which is not conducive to the absorption and loss of electromagnetic energy. Figure 5e–h shows the 2D color mapping of the $\left|Z_{in}/Z_{0}\right|$ values of the BC samples at a frequency of 2–18 GHz. It can be seen that BC-3 and BC-4 have relatively excellent area ratios in the range of 0.8–12.0. This means that they have the best impedance-matching characteristics. Impedance-matching characteristics are crucial in electromagnetic wave-absorbing materials. Good impedance matching can minimize electromagnetic wave reflection, increase absorption efficiency, and achieve wideband electromagnetic wave absorption. Multiple loss mechanisms (dielectric loss, magnetic loss, interface polarization, eddy current effect, etc.) enhance the absorption performance simultaneously. At the same time, there are multiple polarization mechanisms within the material (such as dipole polarization and interface polarization), interface effects between magnetic and dielectric materials, multi-scale and multi-layer subporous structures in bamboo-based carbon/Co/CoO composites, and internal defects in the composite phase structure of the material. These mechanisms have a positive promoting effect on the absorption efficiency of electromagnetic waves in different frequency bands, thereby enabling the material to achieve excellent impedance matching. Therefore, BC-3 achieved satisfactory electromagnetic loss characteristics.

To further elucidate the electromagnetic wave absorption effect of the BC series samples in practical life, we simulated the loss of electromagnetic waves generated by outdoor far-field radar at different angles using CST (CST 2020 Studio Suite) electromagnetic simulation. In CST electromagnetic simulation, the key parameters of electromagnetic wave-absorbing materials are the complex dielectric constant and the complex magnetic permeability of the material. They affect the propagation and absorption of electromagnetic waves in materials. To ensure the prevention of electromagnetic wave transmission affecting the experimental results, PEC (perfect conductor conductor) material with dimensions of 100 mm × 100 mm × 2.0 mm (perfect conductor, capable of completely reflecting electromagnetic waves) was added to the back of the sample (Appendix A). PEC perfect conductor material can serve as the ideal reflection boundary for the bottom layer of the model material, preventing the projection of electromagnetic waves and affecting the simulation experiment results. In addition, the excitation source is set as a plane-wave excitation because it can simulate the propagation of electromagnetic waves under far-field conditions, provide uniform illumination, and is suitable for analyzing electromagnetic wave scattering at different angles. The simulation model adopts a symmetrical model, which can better simulate uniform scattering from multiple angles and efficiently present the simulation. Simulate the electromagnetic loss performance of BC series samples by replacing the coating material on the PEC surface. Specifically, set the coating thickness of BC-3 to 2.8 mm. The calculation frequency is set to 8.2 GHz. The simulated electromagnetic wave incidence angle range is set to −180° to 180°. At the same time, the Z-axis direction (detection angle of 0°) is defined as the incident direction of electromagnetic waves. The 3D electromagnetic scattering energy cluster appearing on the surface of the BC-3 sample is the smallest, and the lowest scattering signal indicates that almost all electromagnetic waves are absorbed by the BC-3 material and the large amount of reflection generated by other samples. It is worth noting that RCS is closely related to the incident angle of electromagnetic waves. The RCS distribution may vary at different angles, especially at certain specific angles (such as vertical or horizontal incidence) where significant changes may occur. Set different incident angles in CST, calculate the characteristic curve of RCS changing with incident angle, and analyze whether the results are consistent with expectations. Furthermore, we present the electromagnetic signals on the y-o-z plane in three-dimensional space in detail through two-dimensional curves. As shown in Figure 6e, when the detection angle is within the range of −60°–60°, the BC-3 sample has the smallest RCS value. This indicates that almost all electromagnetic energy incidents on the material surface in space are effectively absorbed. Figure 6f shows the difference in RCS between the BCs series samples and PEC conductor materials at different detection angles (0° to 80°). It can be seen that the BC-3 sample reaches a maximum of 33.3 dB m^2^ at a detection angle of 10°. When electromagnetic waves radiate on the surface of materials at an incident angle of 0 degrees, they are more prone to electromagnetic reflection, which is not conducive to absorption. It should be noted that the BC-3 sample still has the maximum RCS attenuation value (24.9 dB m^2^) at a detection angle of 0°. It shows an excellent electromagnetic wave absorption effect. By observing the two-dimensional RCS images at Phi of 90° and detection angle range of −180° < 0 < 180°, it can be seen that the BC-3 sample exhibits excellent wide-angle electromagnetic absorption performance. It can absorb electromagnetic radiation energy at most incident angles in space. In bamboo-based carbon/Co/CoO multi-interface heterojunction composite materials, carbonized bamboo and Co/CoO magnetic nanoparticles can be described by an RLC equivalent circuit model. With the increase of Co/CoO concentration from BC-1 to BC-3, conductivity is enhanced, resulting in a decrease in resistance (R) and an increase in electromagnetic wave absorption. The interface polarization decreases, and the capacitance (C) decreases at high concentrations, but the dielectric loss still increases. Magnetic response is enhanced, inductance (L) increases, and magnetic loss dominates electromagnetic wave absorption. The increase in concentration has led to a significant improvement in the electromagnetic wave absorption performance of the material, especially under the synergistic effect of magnetic loss and dielectric loss, which enhances the wideband absorption performance of the material towards electromagnetic waves (Appendix A). In summary, the construction of an anisotropic heterojunction interface with magnetic and dielectric coupling has been achieved based on a periodic matrix multi-layer array repeating unit with a honeycomb-like structure. The multi-interface structure of heterojunctions causes multiple scattering and reflection of electromagnetic waves at the interfaces of different phases when propagating inside the multi-level porous structure. The introduction of magnetic nanoparticles forms rich heterojunction interfaces with the carbon matrix, enhancing dipole polarization and interface polarization effects. The electromagnetic wave absorption performance of the material has been significantly improved through interface polarization, enhanced multiple scattering and reflection effects, synergistic effects of magnetic and dielectric properties, and local electric field distortion (Figure 7).

## 4. Conclusions

In this work, we report a bamboo-based carbon/Co/CoO heterojunction structure based on a multi-layer periodic matrix array for efficient electromagnetic wave absorption. By oxidizing and removing lignin components from bamboo, the internal multi-level pore structure is enriched, and the surface-rich functional chemical groups are directionally regulated, promoting the full infiltration of cobalt–ion solution. Furthermore, thermal decomposition leads to the formation of rich heterogeneous structures within the material. The coupling effect of magnetism and dielectric at the interface promotes charge accumulation and separation, leading to strong interface polarization and relaxation phenomena. The synergistic effect enhances the dielectric loss and hysteresis loss of the material, making the material have excellent absorption performance for electromagnetic waves in a wide frequency range, simultaneously promoting the formation of optimal impedance matching. Therefore, by adjusting the appropriate impregnation concentration, the reflection loss RL value of the bamboo-based carbon/Co/CoO composite material reached −54.7 dB (BC-3). In addition, the effective absorption bandwidth of BC-3 reached 5.1 GHz at a matching thickness of 1.55 mm. The electromagnetic simulation results of CST far-field radar show that the material exhibits excellent RCS attenuation of 33.3 dB·m^2^ in the detection angle range of −90° < 0 < 90°. This work provides new directions for the diversified development of green biomass and the optimization design of magnetic and dielectric coupling in periodic array structures.

## Figures and Tables

**Figure 1 materials-17-05239-f001:**
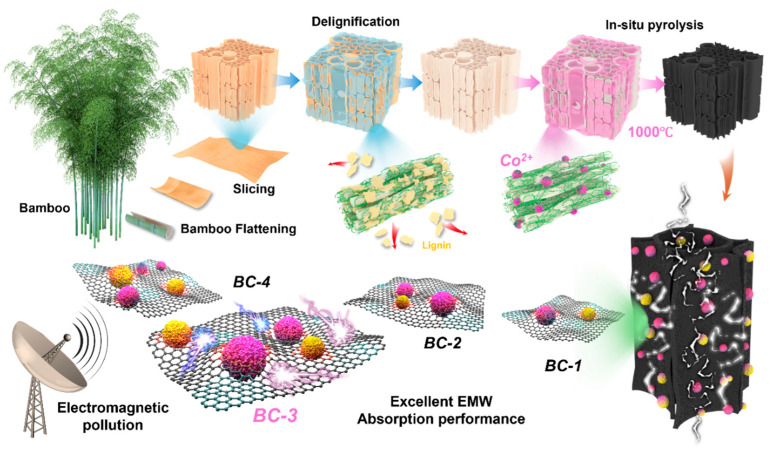
Schematic diagram of the preparation route of bamboo-based carbon/Co/CoO multi-interface heterojunction absorber based on multi-layer periodic matrix array.

**Figure 2 materials-17-05239-f002:**
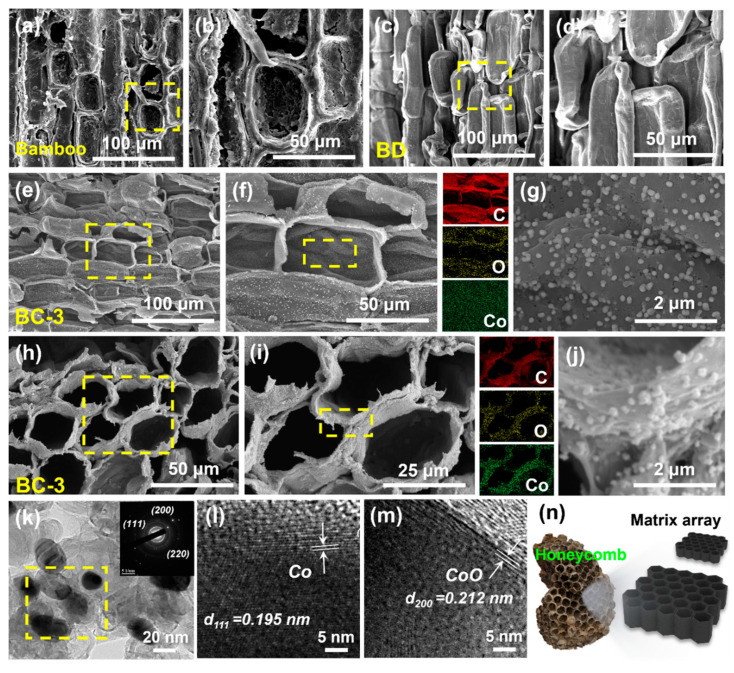
(**a**–**d**) SEM images of original bamboo and bamboo slices after lignin removal. (**e**–**j**) SEM and EDS images of the radial and transverse cross-sections of BC-3, as well as locally magnified SEM images. HRTEM image and selective region electron diffraction pattern of (**k**–**m**) BC-3. (**n**) Schematic diagram of the honeycomb-like multi-layer matrix array structure inside bamboo.

**Figure 3 materials-17-05239-f003:**
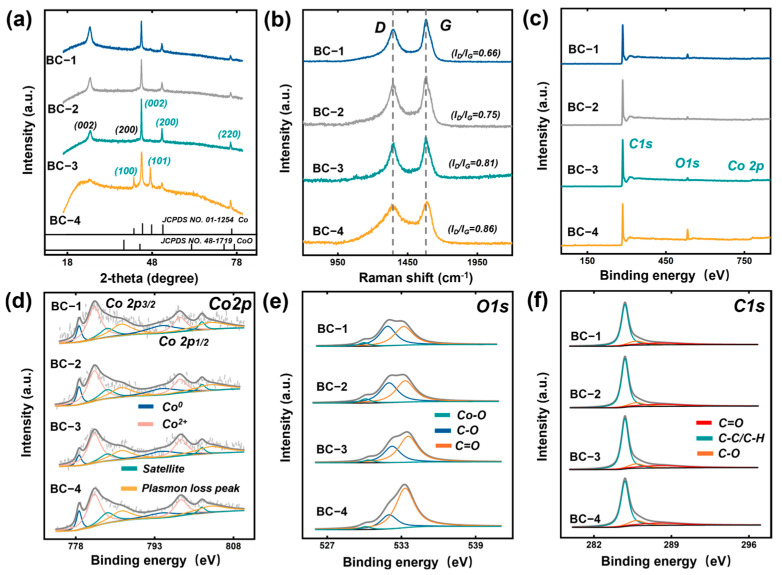
(**a**,**b**) XRD and Raman spectra curves of BCs. (**c**) XPS survey and (**d**–**f**) high-resolution XPS spectra (Co2p, C1s, and O1s).

**Figure 4 materials-17-05239-f004:**
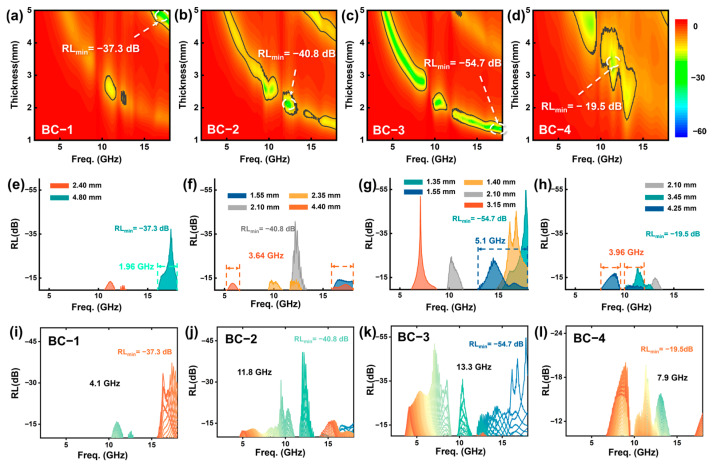
(**a**–**d**) Two-dimensional color-filled mapping images of RL values with frequencies ranging from 2.0 GHz to 18.0 GHz and thicknesses ranging from 1.00 mm to 5.00 mm. (**e**–**h**) Representative RL curves of BCs samples at corresponding matching thicknesses in the frequency range of 2–18 GHz. Representative RL curves of effective absorption performance of (**i**–**l**) BCs samples at different matching thicknesses.

**Figure 5 materials-17-05239-f005:**
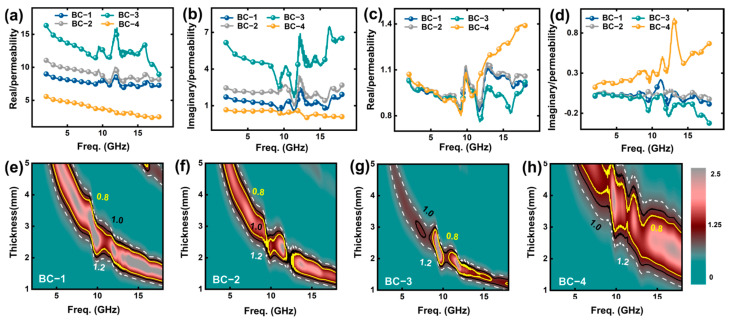
(**a**–**d**) $\varepsilon^{\prime }$ and $\varepsilon^{\prime \prime }$ curves. Two-dimensional color mapping image of the $\left|Z_{in}/Z_{0}\right|$ value of (**e**–**h**) BCs.

**Figure 6 materials-17-05239-f006:**
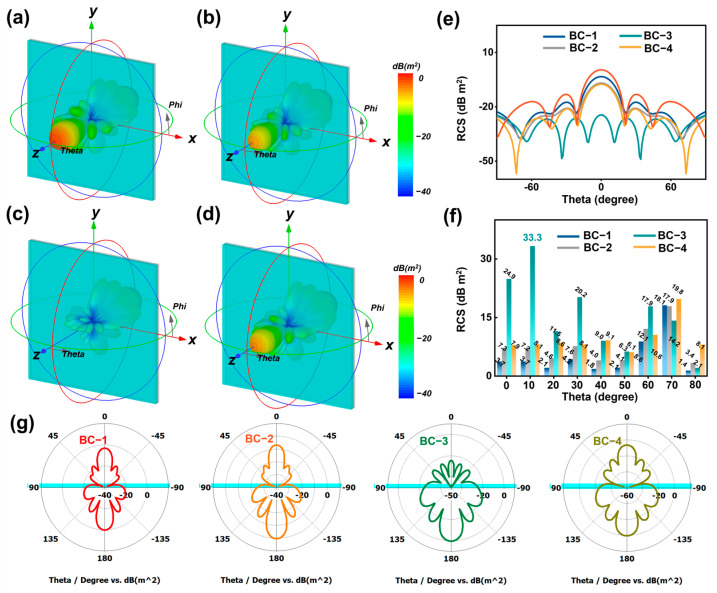
(**a**–**d**) Schematic diagram of CST electromagnetic simulation of BC series samples. (**e**) RCS value curve of BC series samples (detection angle −90° < 0 < 90°). (**f**) RCS attenuation value image of BC series samples. (**g**) RCS value curve (detection angle −180° < 0 < 180°).

**Figure 7 materials-17-05239-f007:**
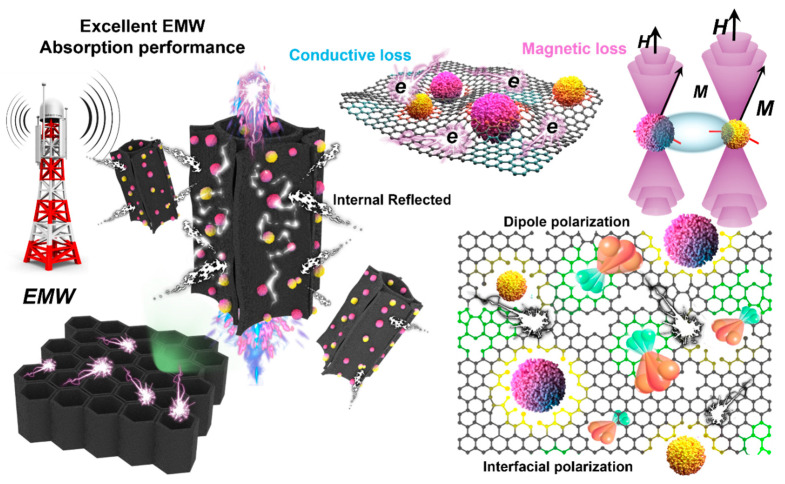
Schematic diagram of EMW absorption mechanism of bamboo-based carbon/Co/CoO composite absorbent.

## Data Availability

The original contributions presented in the study are included in the article, further inquiries can be directed to the corresponding authors.

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
