# Peer review of "Bamboo-Based Carbon/Co/CoO Heterojunction Structures Based on a Multi-Layer Periodic Matrix Array Can Be Used for Efficient Electromagnetic Attenuation"

_materials, 2024, doi:10.3390/ma17215239_

Round 1

Reviewer 1 Report

Comments and Suggestions for Authors

The paper is devoted for bamboo-based carbon heterojunction structure based on multi-layer periodic matrix array investigations for electromagnetic applications. The topic is generally interesting, however the paper contain unexplained places (below) and need major revisions.

Lines 380-381, the sentence ‘’multiple semicircles (Cole-Cole semicircles) appear in the ε'- ε'' curve of the BCs sample, indicating rich multipolar relaxation inside the material’’ is not correct, measurements were performed in narrow frequency range (2-18 GHz) and in this frequency range can be observed only one relaxation (please look on Debye equation). Therefore Cole-Cole pictures presented in Fig. 5 (i), (j), (k), (l) are also not informative.

What is relation between your samples structure presented in Fig. 2 and dielectric and microwave properties presented in Figs. 4-5? What is an impact of Co and CoO concentration on dielectric and microwave properties presented in Figs. 4-5?

Lines 335-336, ‘’excellent interface effect’’ what you mean there?

Numbers and measurements units should be written separately, for example line 129 it should be ‘’4 cm’’.

Conclusions should be rewritten in more consistent and informative way.

Please compare your results with results presented in [1-3].

[1] P. Bertašius et al, , Polymers 15, 1053 (2023).

 [2] M. Bleija et al, Polymers 15, 515 (2023).

[3] M. Bleija et al, Nanomaterials 12, 3671 (2022).

Author Response

Dear Editor:

We are grateful for your correspondence and the valuable comments from the reviewer on the quality of our manuscript. We would like to thank the reviewer for providing his concerns, which will further strengthen the presentation of our work. Below we respond to the comments from you and the reviewers of our manuscript in a point-by-point manner.

Reviewer #1:

The paper is devoted for bamboo-based carbon heterojunction structure based on multi-layer periodic matrix array investigations for electromagnetic applications. The topic is generally interesting, however the paper contain unexplained places (below) and need major revisions.

Response: We greatly appreciate the insightful and constructive comments and suggestions provided by the reviewers. We believe that the input of the reviewers enabled us to complete a significantly improved manuscript. In response to the comments of the reviewers, we have made extensive revisions to the writing, presentation, and analysis of the research results. These opinions and suggestions not only enable us to provide a highly improved manuscript, but also inspire us to conduct more in-depth research on it in future work. Meanwhile, we also hope that the relevant modifications can meet the requirements of the reviewers.

1) Lines 380-381, the sentence ‘’multiple semicircles (Cole-Cole semicircles) appear in the ε'- ε'' curve of the BCs sample, indicating rich multipolar relaxation inside the material’’ is not correct, measurements were performed in narrow frequency range (2-18 GHz) and in this frequency range can be observed only one relaxation (please look on Debye equation). Therefore Cole-Cole pictures presented in Fig. 5 (i), (j), (k), (l) are also not informative.

Response: We greatly appreciate the insightful and constructive comments and suggestions provided by the reviewers. Based on the reviewer's comments, we have carefully reviewed and organized the corresponding sections of the manuscript. We deeply apologize for the incomplete description of the content of the Cole Cole semicircle drawn for material polarization loss. Meanwhile, we also hope that the relevant modifications can meet the requirements of the reviewers. Specifically, as follows:

As mentioned by the reviewer, in this manuscript, we removed lignin components that hinder impregnation diffusion inside bamboo through oxidation modification, while selectively adjusting the abundant functional chemical groups on the surface of cellulose and hemicellulose. A series of bamboo-based carbon/Co/CoO multi interface heterojunction electromagnetic absorbers with multi-layer periodic matrix arrays were obtained by introducing cobalt containing solutions of different concentrations into bamboo through vacuum impregnation and in-situ pyrolysis.

Among them, we analyzed the polarization loss inside the material through the relationship diagram () between the real and imaginary parts of the dielectric constant. Multiple semicircles (Cole Cole semicircles) appeared in the - curve of the sample. As stated by the reviewer, the measurement was conducted within a narrow frequency range (2-18GHz), where only one relaxation can be observed. In the frequency range of 2-18 GHz, if multiple semicircles (i.e. multiple Cole Cole semicircles) appear on the - curve of the absorbing material, it may indicate the presence of multiple complex polarization mechanisms or composite effects inside the material. Although a single relaxation phenomenon is usually associated with a semicircle, in some cases, multi-polarization mechanisms may occur simultaneously within that frequency range, resulting in the appearance of multiple semicircles. The specific content is as follows:

The combined effect of multiple polarization mechanisms: There may be multiple different polarization mechanisms inside the material, which work simultaneously in the frequency range of 2-18 GHz. Mainly including: (1) Dipole polarization: Due to the action of electromagnetic waves, dipoles are repositioned inside the material, resulting in relaxation effects. Dipole polarization can cause molecular or atomic dipoles in materials to undergo orientation changes under external electric fields, during which energy is dissipated in the form of heat, thereby increasing dielectric loss and enhancing electromagnetic wave absorption. Meanwhile, dipole polarization not only increases dielectric loss at the microscopic level, but also generates multiple scattering effects within the material. When the dipole distribution in the material is uneven or there are heterogeneous structures such as interfaces and defects, electromagnetic waves will be repeatedly scattered and reflected, increasing the absorption path of electromagnetic waves and further improving the absorption performance. (2) Interface polarization: Different phase interfaces of materials (such as between metal nanoparticles and dielectric matrix) may induce interface polarization in a lower frequency range, which is commonly seen in composite materials. Interface polarization usually occurs at the interface of heterostructures or multiphase materials. In these interface regions, there are differences in the conductivity and dielectric constant of different materials, which prevent charges from freely passing through the interface and accumulate at the interface, forming polarization. The process of charge accumulation and polarization is accompanied by energy dissipation, which is converted into thermal energy, thereby increasing dielectric loss and promoting electromagnetic wave absorption. Interface polarization effectively improves the performance of electromagnetic wave absorbing materials by enhancing dielectric loss, improving impedance matching, increasing scattering and reflection paths, and broadening absorption frequency bands. Especially in heterostructures and composite materials, interface polarization significantly enhances material absorption efficiency. (3) Electronic polarization and ion polarization: These polarization mechanisms may work in a higher frequency range. Electron polarization and ion polarization can form a synergistic effect in bamboo-based carbon multiphase composites. Electronic polarization is more suitable for absorbing high-frequency electromagnetic waves, while ion polarization exhibits significant absorption effects in the low-frequency range. By designing a reasonable material structure, these two polarization mechanisms can work together to achieve wideband and efficient absorption of electromagnetic waves. In addition, both can enhance the overall absorption performance of the material by enhancing interface polarization, improving dielectric loss, and multiple scattering mechanisms.

Multi-scale structure of materials: In this manuscript, bamboo-based carbon/Co/CoO multi-interface heterojunction electromagnetic absorbers have multi-scale composite structures (such as Co/CoO nanoparticles, multi-level pore structures inside bamboo, between Co and carbon matrix, and between CoO and carbon matrix, etc.), and structures of different scales may produce different polarization effects in different frequency ranges. Each structural feature, such as the size of Co/CoO nanoparticles, the rich phase interfaces between Co and carbon matrix, CoO and carbon matrix, and Co/CoO nanoparticles, as well as the numerous defects present within the material, may trigger an independent relaxation process. Therefore, multiple Cole Cole semicircles may be related to the different scales and structures of bamboo-based carbon/Co/CoO composites.

Complex interface effects: In bamboo-based carbon/Co/CoO composite materials, the construction of anisotropic heterojunction interfaces with magnetic and dielectric coupling has been achieved based on periodic matrix multilayer array repeating units resembling honeycomb structures. The multi-interface structure of heterojunctions causes multiple scattering and reflection of electromagnetic waves at the interfaces of different phases when propagating inside the multi-level porous structure. The introduction of magnetic nanoparticles forms rich heterojunction interfaces with the carbon matrix, enhancing dipole polarization and interface polarization effects. Due to the electromagnetic non-uniformity between different components or phase interfaces within the material, Maxwell Wagner Sillars polarization effects may occur. This effect occurs at material interfaces with significant differences in dielectric constant and conductivity, which may lead to multiple relaxation phenomena, resulting in multiple semicircles. Especially when there are multiple interacting interfaces in the material (such as Co/CoO metal nanoparticles embedded in carbonized bamboo dielectric matrix). In this manuscript, we constructed an anisotropic heterojunction interface with magnetic and dielectric coupling based on a honeycomb-like periodic matrix multilayer array repeating unit. The lignin component of bamboo was removed by oxidation, enriching the impregnation pores and uniform adsorption sites of the magnetic medium. Further in-situ pyrolysis promotes the formation of a large number of electric dipoles at the interface between the magnetic medium and dielectric coupling inside the periodic cell carbon skeleton, enhancing interface polarization and relaxation. These interface effects will generate multiple relaxation processes.

Figure 5. (a-d)  and  curves. 2D color mapping image of the  value of (e-h) BCs. (i-l) Cole-Cole curve of BCs.

Multiple relaxation caused by impurities or defects within the composite phase structure of materials: Impurities or structural defects within the material (such as pores, grain boundaries, dislocations, etc.) may also cause multiple relaxation processes. These structural defects can cause charges to accumulate and release at different positions, forming multiple relaxation mechanisms and appearing as multiple semicircles on the complex dielectric constant curve. For composite materials with multiple phase structures (such as conductive and non-conductive phases), the combination of different phases can lead to different electromagnetic properties, and different phases may produce independent relaxation processes in different frequency ranges. For example, the conductive phase in the carbon matrix may exhibit significant relaxation at higher frequencies, while the polarization relaxation of the non-conductive phase may play a role in the lower frequency range.

In summary, although theoretically only one major relaxation phenomenon usually occurs in the frequency range of 2-18 GHz, the appearance of multiple Cole-Cole semicircles may be attributed to the multi-polarization mechanism inside the material, interface effects, complexity of composite structures, multi-scale structures, and multiple relaxations caused by impurities or defects inside the composite phase structure of the material. Multiple Cole semicircles were formed. This further intuitively demonstrates the polarization relaxation process in the material. Magnetic carbon composite materials typically involve multiple polarization mechanisms, such as electronic polarization, ion polarization, dipole polarization, and interface polarization. The composite of magnetic components in materials and carbon-based materials often exhibits complex electromagnetic behavior, manifested as multiple polarization relaxation processes. By examining the shape and number of Cole-Cole semicircles, it is possible to determine whether the multiple polarization mechanisms work together, and further determine the polarization characteristics of each component and their contribution to electromagnetic wave absorption. Multiple semicircles may indicate the presence of multiple polarization mechanisms within the material, and these mechanisms play different roles in absorbing electromagnetic waves at different frequencies. The multi polarization mechanism in materials helps to achieve wideband absorption, thereby improving the overall absorption effect.

Correspondingly, in the manuscript, we have made corresponding supplementary modifications as follows:

In Line 421-431 on Page 11

“As shown in Figure 5i-l, multiple semicircles (Cole-Cole semicircles) appear in the - curve of the BCs sample, indicating rich multipolar relaxation inside the material, which has a positive promoting effect on dielectric loss.” is revised as “As shown in Figure 5i-l, multiple semicircles (Cole Cole semicircles) appear in the - curve of the BC sample. Usually, only one major relaxation phenomenon occurs in the frequency range of 2-18 GHz, but the appearance of multiple Cole Cole semicircles may be attributed to the multi-polarization mechanism within the material (dipole polarization and interface polarization, etc.), interface effects between magnetic and dielectric media, multi-scale and multi-level pore structures in bamboo-based carbon/Co/CoO composites, and internal defects in the composite phase structure of the material, which form multiple Cole-Cole semicircles. Furthermore, the multi-polarization mechanism helps to achieve wideband absorption, and these mechanisms have a positive promoting effect on electromagnetic wave absorption at different frequencies. Thereby improving the overall electromagnetic wave absorption effect.

2) What is relation between your samples structure presented in Fig. 2 and dielectric and microwave properties presented in Figs. 4-5? What is an impact of Co and CoO concentration on dielectric and microwave properties presented in Figs. 4-5?

Response: We greatly appreciate the insightful and constructive comments and suggestions provided by the reviewers. Based on the reviewer's comments, we have carefully reviewed and organized the corresponding sections of the manuscript. Meanwhile, we also hope that the relevant modifications can meet the requirements of the reviewers. Specifically, as follows:

As mentioned by the reviewer, in this manuscript, we report a bamboo-based carbon/Co/CoO heterojunction structure based on a multi-layer periodic matrix array for efficient electromagnetic wave absorption. By thoroughly immersing bamboo slices that have undergone oxidation modification treatment with appropriate cobalt salt concentration. Based on a periodic matrix multilayer array unit cell resembling a honeycomb, a rich multiphase heterojunction structure was formed under the action of high-temperature pyrolysis. The coupling effect of magnetism and dielectric promotes the formation of a large number of electric dipoles between multiple interfaces, enhancing interface polarization and relaxation effects. The local carrier traps and uneven electromagnetic density formed by local electric field distortion enhance the dielectric and hysteresis losses of the material, promoting the formation of optimal impedance matching.

Bamboo has a shorter growth cycle and more abundant resources. Compared to other carbon-based materials, bamboo is composed of regularly arranged thin-walled cells and vascular bundles, forming a highly ordered honeycomb-like multi-layer matrix array microstructure. This structure endows bamboo-based carbon materials with good mechanical strength and low density. The honeycomb-like microstructure itself can provide materials with a natural multi-level pore structure, enriching the multiple reflection and scattering effects of electromagnetic waves. In a multi-layer periodic matrix array based on bamboo carbon heterojunction structure, the relationship between the repeated unit structure and the dielectric and microwave properties of the material involves multiple mechanisms. The design of this structure and the composition of the materials will affect the propagation, reflection, and absorption behavior of electromagnetic waves.

Specifically, Figure 2 shows the repetitive unit structure of the periodic matrix array inside bamboo, and the bamboo has a large number of multi-level pores inside. The removal of lignin components from bamboo through oxidation enriches the impregnation pores and uniform adsorption sites of the magnetic medium. The periodic structure inside bamboo can form electromagnetic band gaps, which prevent electromagnetic waves in certain frequency ranges from passing through, thereby improving its absorption performance. At certain frequencies, under the influence of appropriate periodic unit sizes and electromagnetic parameters of the material, local resonance may occur inside the material, which enhances the absorption of electromagnetic waves. The size and thickness of periodic units can be matched with the wavelength of electromagnetic waves at a certain frequency, and through optimized design, their absorption effect can be significantly enhanced within a specific frequency range. (2) Within the bamboo-based carbon/Co/CoO composite material, an anisotropic heterojunction interface with magnetic and dielectric coupling was constructed based on a periodic matrix multi-layer array repeating unit resembling a honeycomb structure. There are interfaces between different phases in heterojunctions, and the electromagnetic response at the interface significantly affects the dielectric properties: Interface polarization (Maxwell Wagner effect): In heterojunctions, the difference in conductivity or dielectric constant at the interface of different materials such as Co, CoO, and carbon matrix can lead to charge accumulation at the interface, forming interface polarization phenomenon. This effect will increase dielectric loss, thereby enhancing the absorption of electromagnetic waves. Multipolarization mechanism: Due to the composite of different phase materials (such as Co/CoO nanoparticle size, the rich phase interfaces between Co and carbon matrix, CoO and carbon matrix, and Co/CoO nanoparticles), multiple polarization mechanisms may occur within the heterojunction structure, including electronic polarization, ion polarization, dipole polarization, and interface polarization. The composite of magnetic components in materials and carbon-based materials often exhibits complex electromagnetic behavior, manifested as multiple polarization relaxation processes. The multi-polarization mechanism in materials helps to achieve wideband absorption, thereby improving the overall absorption effect. (3) The dielectric constant of materials and microwave absorption: The real and imaginary parts of the dielectric constant (ε) represent the energy storage capacity and loss capacity of the material, respectively. High dielectric loss materials are more conducive to converting electromagnetic wave energy into thermal energy, thereby improving microwave absorption performance. The bamboo-based carbon/Co/CoO heterojunction structure promotes the enhancement of dielectric constant. The superposition effect of the relative permittivity of different materials in a heterojunction will increase the overall permittivity of the composite material, enhancing its response to electromagnetic waves. Due to the high conductivity and good dielectric loss ability of bamboo carbon structure, it can effectively dissipate electromagnetic wave energy.

Figure 2. (a-d) SEM images of original bamboo and bamboo slices after lignin removal. SEM and EDS images of the radial and transverse cross-sections of BC-3, as well as locally magnified SEM images. HRTEM image and selective region electron diffraction pattern of (k-m) BC-3. (n) Schematic diagram of the honeycomb like multi-layer matrix array structure inside bamboo.

Meanwhile, in this manuscript, different concentrations of cobalt-containing solutions were introduced into bamboo through vacuum impregnation, and a series of bamboo-based carbon/Co/CoO multi-interface heterojunction electromagnetic absorbers with multi-layer periodic matrix arrays were obtained through in-situ pyrolysis. The change in concentration results in different magnetic medium ratios, defect concentrations, and dielectric properties. Further research has shown that magnetic media and dielectric coupling interfaces can cause charge redistribution and form electric dipoles under the action of electromagnetic fields, further enhancing interface polarization and relaxation. The formation of local carrier traps and uneven electromagnetic energy density at heterojunction interfaces enhances local dielectric losses and eddy current effects. By optimizing the heterojunction structure, excellent magnetic loss and dielectric coupling effects are constructed to obtain suitable impedance matching characteristics.

Figure 4. (a-d) 2D color-filled mapping images of RL values with frequencies ranging from 2.0 GHz to 18.0 GHz and thicknesses ranging from 1.00 mm to 5.00 mm. (e-h) Representative RL curves of BCs samples at corresponding matching thicknesses in the frequency range of 2-18 GHz. Representative RL curves of effective absorption performance of (i-l) BCs samples at different matching thicknesses.

Figure 5. (a-d)  and  curves. 2D color mapping image of the  value of (e-h) BCs. (i-l) Cole-Cole curve of BCs.

Specifically, (1) Enhancement of magnetic and electromagnetic properties of materials: Co and CoO have good magnetism, and their introduction will enhance the magnetic permeability (μ) of the material, thereby affecting its microwave absorption performance. The introduction of Co and CoO will enhance the magnetic loss of the material, especially in the high-frequency range (such as microwave frequency). This magnetic loss is related to the magnetization reversal and eddy current loss inside the material, which helps to absorb the magnetic field components of electromagnetic waves. At the same time, the presence of Co and CoO may trigger local magnetic resonance absorption effects, enhancing the material's absorption performance at specific frequencies. (2) The effect of Co and CoO concentration on dielectric constant: At low concentrations, Co or CoO at low concentrations usually only slightly enhances the dielectric and magnetic properties of the material, and the dielectric constant and loss are relatively stable. This is because at low concentrations, the microstructure distribution of cobalt and cobalt oxide is sparse, which cannot significantly change the electromagnetic properties of the material. At high concentrations: With the increase of Co and CoO concentrations, the polarization mechanism and magnetic loss inside the material are significantly enhanced, and the imaginary part of the dielectric constant increases, indicating an increase in the dielectric loss of the material. High concentrations of cobalt increase the conductivity inside the material, resulting in more energy loss when electromagnetic waves pass through the material. Simultaneously introducing a multi-polarization relaxation mechanism into the material, resulting in multiple relaxation processes that affect the frequency dependence of the complex dielectric constant. (3) High concentrations of Co and CoO may cause particles to aggregate in the material, forming large-scale conductive networks or clusters. This cluster structure may suppress interface polarization and dipole polarization, reduce the energy storage and loss capacity of the material, and lead to a decrease in dielectric constant. As the concentration of Co and CoO increases, the coupling effect between magnetic particles strengthens, resulting in an increase in the real and imaginary parts of magnetic permeability, especially in eddy current and hysteresis losses, which are more significant at high concentrations.

In summary, the repetitive unit structure of periodic matrix arrays affects the microwave absorption performance of materials through electromagnetic bandgap effects, resonance effects, and geometric size and frequency matching. The interface polarization effect and multi-polarization mechanism in heterojunctions enhance the dielectric loss and absorption performance of materials. In addition, the introduction of Co and CoO enhances the microwave absorption performance of the material through magnetic loss and resonance absorption. As the concentration increases, the dielectric constant and magnetic loss of the material gradually increase, and the coupling effect between magnetic particles is enhanced, resulting in an increase in the real and imaginary parts of the magnetic permeability, especially eddy current loss and hysteresis loss, which are more significant at high concentrations. However, excessive concentration may cause excessive conductivity loss, and the conductivity loss and particle aggregation effect suppress polarization, leading to a decrease in dielectric constant. On the contrary, it suppresses the absorbing effect of the material.

Correspondingly, in the manuscript, we have made corresponding supplementary modifications as follows:

In Line 325-330 on Page 9

 “The repetitive unit structure of the periodic matrix array inside bamboo can form an electromagnetic bandgap, which prevents electromagnetic waves in certain frequency ranges from passing through, thereby improving the absorption performance. At certain frequencies, under the influence of appropriate periodic unit sizes and electromagnetic parameters of the material, local resonance may occur inside the material, which enhances the absorption of electromagnetic waves.

In Line 374-377 on Page 10

“High concentrations of Co and CoO may cause particles to aggregate in the material, forming large-scale conductive networks or clusters. This cluster structure may suppress interface polarization and dipole polarization, reduce the energy storage and loss capacity of the material, and lead to a decrease in dielectric constant.

In Line 396-401 on Page 10

With the increase of Co and CoO concentrations, the coupling effect between magnetic particles is enhanced, especially the eddy current loss and hysteresis loss are more significant at high concentrations. However, excessive concentration may cause conductivity loss and particle aggregation effects, which suppress polarization and lead to a decrease in dielectric constant. On the contrary, it suppresses the absorbing effect of the material.

3) Lines 335-336, ‘’excellent interface effect’’ what you mean there?

Response: We greatly appreciate the insightful and constructive comments and suggestions provided by the reviewers. Based on the reviewer's comments, we have carefully reviewed and organized the corresponding sections of the manuscript. Meanwhile, we also hope that the relevant modifications can meet the requirements of the reviewers. Specifically, as follows:

As mentioned by the reviewer, in this manuscript, we removed lignin components that hinder impregnation diffusion inside bamboo through oxidation modification, while selectively adjusting the abundant functional chemical groups on the surface of cellulose and hemicellulose. A series of bamboo-based carbon/Co/CoO multi-interface heterojunction electromagnetic absorbers with multi-layer periodic matrix arrays were obtained by introducing cobalt-containing solutions of different concentrations into bamboo through vacuum impregnation and in-situ pyrolysis. The change in concentration results in different magnetic medium ratios, defect concentrations, and dielectric properties. Based on a periodic matrix multilayer array unit cell resembling a honeycomb, a rich multiphase heterojunction structure was formed under the action of high-temperature pyrolysis. The coupling effect of magnetism and dielectric promotes the formation of a large number of electric dipoles between multiple interfaces, enhancing interface polarization and relaxation effects.

The excellent interface polarization effect specifically refers to the polarization phenomenon caused by the accumulation or retention of charges at different phase interfaces in bamboo-based carbon/Co/CoO multi-interface heterojunction materials. This effect originates from the differences in electromagnetic properties between different phases within the material. The dielectric constant or conductivity between different phase structures within the material varies, leading to the accumulation of charges at the interface, thereby enhancing the dissipation and conversion of electromagnetic wave energy, manifested as an improvement in the material's ability to absorb electromagnetic waves. Specifically, as shown below:

(1) In multiphase composite structures such as bamboo-based carbon/Co/CoO heterojunction materials, different phase materials such as bamboo-based carbon, Co, CoO, etc. have significant differences in dielectric and conductivity at the interface. When electromagnetic waves act on these heterogeneous interfaces, charges are trapped or aggregated at the interfaces, forming local polarization phenomena. Due to the difficulty of free flow of charges between bamboo-based carbon, cobalt metal, and cobalt oxide materials across the interface, the accumulation of charges at the interface forms dipoles, which generate dynamic polarization effects with the periodic changes of electromagnetic waves.

(2) The multi-interface heterojunction structure formed by bamboo-based carbon and Co/CoO provides a large number of interfaces, and the charge accumulation on these interfaces enhances the overall polarization ability. Multiple interfaces and uniform distribution: Due to the removal of lignin through oxidation, abundant pores and adsorption sites are generated inside bamboo, allowing cobalt-containing solutions to be evenly distributed and penetrate into the carbon skeleton, thereby forming a uniformly distributed interface between Co/CoO nanoparticles and bamboo-based carbon after pyrolysis. Charge accumulation and dipole polarization promote rental. On these heterogeneous interfaces, due to differences in dielectric constant and conductivity, a large amount of charge accumulation occurs, resulting in significant dipole polarization effects. These dipoles respond to electromagnetic waves, forming strong interface polarization effects.

(3) The periodic matrix multilayer array structure resembling a honeycomb not only increases the surface area of the material, but also increases the number of heterogeneous interfaces in the material, making the interface polarization effect more significant. The advantages of periodic structures: Periodic structures allow electromagnetic waves to reflect and propagate multiple times in materials, increasing the opportunities for interaction between electromagnetic waves and interfaces and enhancing interface polarization. Anisotropic interface effect: The multi-interface heterojunction of this material exhibits anisotropy, meaning that the interface properties and polarization effects differ in different directions, increasing the material's response to different components of electromagnetic waves and further improving the absorption effect.

(4) By introducing Co and CoO nanoparticles into a cobalt-containing solution, a coupling interface between strong magnetic particles and dielectric materials is formed inside the material. This coupling effect between magnetism and dielectric materials enhances the interface polarization effect. Magnetic and dielectric synergistic effect: Co and CoO nanoparticles, as magnetic materials, form a synergistic effect with bamboo-based carbon. The coupling of magnetic and dielectric materials enhances the loss ability of electromagnetic waves, especially the resonance effect on the magnetic field components in electromagnetic waves, further improving the absorption ability. The synergistic effect of multiple absorption mechanisms: In addition to interface polarization effect, there are also multiple absorption mechanisms such as magnetic loss and eddy current loss in the material. These mechanisms work together to enable the material to efficiently absorb and dissipate electromagnetic wave energy.

In summary, the excellent interface polarization effect specifically refers to the formation of a large amount of charge accumulation at the interface in bamboo-based carbon/Co/CoO multi-interface heterojunction materials due to the differences in dielectric constant and conductivity between different phases, resulting in strong dipole polarization and relaxation effects. These effects collectively enhance the material's response to electromagnetic waves, increase dielectric and magnetic losses, and endow the material with excellent electromagnetic wave absorption performance. This excellent interface polarization effect is the result of the combined effects of multi-interface heterojunctions, periodic array structures, and magnetic dielectric coupling effects.

Correspondingly, in the manuscript, we have made corresponding supplementary modifications as follows:

In Line 364-368 on Page 10

Therefore, compared to other samples, BC-3 exhibits excellent interface polarization effect and can effectively dissipate electromagnetic energy through dielectric loss.” is revised as “Therefore, compared to other samples, the rich heterojunction structure inside BC-3 leads to the accumulation and separation of charges at the interface, resulting in strong polarization effects and wideband relaxation phenomena. The synergistic effect of interface polarization and relaxation phenomenon enhances the dielectric loss of the material. Make the material have excellent absorption performance for electromagnetic waves in a wide frequency range.

4) Numbers and measurements units should be written separately, for example line 129 it should be ‘’4 cm’’.

Response: We greatly appreciate the insightful and constructive comments and suggestions provided by the reviewers. Based on the reviewer's comments, we have carefully reviewed and organized the corresponding sections of the manuscript. We deeply apologize for some formatting errors. We have carefully examined the entire text regarding this. Meanwhile, we also hope that the relevant modifications can meet the requirements of the reviewers. Specifically, as follows:

In Line 152 on Page 4

4 cm × 4 cm × 1 mm

In Line 158 on Page 4

Co(NO3)2 ·6H2O

In Line 319-320 on Page 8

As shown in Figure 4e-h, the optimal effective absorption bandwidth of the BC samples are 1.96 GHz (4.8 mm), 3.64 GHz (2.35 mm), 5.1 GHz (1.55 mm), and 3.96 GHz (4.25 mm), respectively.

5) Conclusions should be rewritten in more consistent and informative way.

Response: We greatly appreciate the insightful and constructive comments and suggestions provided by the reviewers. Based on the reviewer's comments, we have carefully reviewed and organized the conclusion section of the manuscript. Meanwhile, we also hope that the relevant modifications can meet the requirements of the reviewers. Specifically, as follows:

As mentioned by the reviewer, in this manuscript, we removed lignin components that hinder impregnation diffusion inside bamboo through oxidation modification, while selectively adjusting the abundant functional chemical groups on the surface of cellulose and hemicellulose. A series of bamboo-based carbon/Co/CoO multi-interface heterojunction electromagnetic absorbers with multi-layer periodic matrix arrays were obtained by introducing cobalt-containing solutions of different concentrations into bamboo through vacuum impregnation and in-situ pyrolysis. In order to better describe and express the conclusion, making it more consistent and informative. We have made adjustments and modifications to the conclusion section of the manuscript.

In Line 514-532 on Page 14

In this work, we report a bamboo-based carbon/Co/CoO heterojunction structure based on a multi-layer periodic matrix array for efficient electromagnetic wave absorption. By thoroughly immersing bamboo slices that have undergone oxidation modification treatment with appropriate cobalt salt concentration. Based on a periodic matrix multilayer array unit cell resembling a honeycomb, a rich multiphase heterojunction structure was formed under the action of high-temperature pyrolysis. The coupling effect of magnetism and dielectric promotes the formation of a large number of electric dipoles between multiple interfaces, enhancing interface polarization and relaxation effects. The local carrier traps and uneven electromagnetic density formed by local electric field distortion enhance the dielectric and hysteresis losses of the material, promoting the formation of optimal impedance matching. Therefore, the minimum RL value of bamboo-based carbon/Co/CoO composite material reaches -54.7 dB (BC-3), and the effective absorption bandwidth is 5.1 GHz (with a matching thickness of only 1.55 mm). The CST electromagnetic simulation results show that the material exhibits excellent RCS attenuation of 33.3 dB · m2 within the detection angle range of -180º<0<180º. This work provides new directions for the diversified development of green biomass and the optimization design of magnetic and dielectric coupling in periodic array structures. ” is revised as “In this work, we report a bamboo-based carbon/Co/CoO heterojunction structure based on a multi-layer periodic matrix array for efficient electromagnetic wave absorption. By oxidizing and removing lignin components from bamboo, the internal multi-level pore structure is enriched, and the surface rich functional chemical groups are directionally regulated, promoting the full infiltration of cobalt ion solution. Furthermore, thermal decomposition leads to the formation of rich heterogeneous structures within the material. The coupling effect of magnetism and dielectric at the interface promotes charge accumulation and separation, leading to strong interface polarization and relaxation phenomena. The synergistic effect enhances the dielectric loss and hysteresis loss of the material. Make the material have excellent absorption performance for electromagnetic waves in a wide frequency range. Simultaneously promoting the formation of optimal impedance matching. Therefore, by adjusting the appropriate impregnation concentration, the reflection loss RL value of the bamboo-based carbon/Co/CoO composite material obtained reached -54.7 dB (BC-3). In addition, the effective absorption bandwidth of BC-3 reached 5.1 GHz at a matching thickness of 1.55mm. The electromagnetic simulation results of CST far-field radar show that the material exhibits excellent RCS attenuation of 33.3 dB·m2 in the detection angle range of -90º<0<90º. This work provides new directions for the diversified development of green biomass and the optimization design of magnetic and dielectric coupling in periodic array structures.

6) Please compare your results with results presented in [1-3].

[1] P. Bertašius et al, Polymers 15, 1053 (2023).

[2] M. Bleija et al, Polymers 15, 515 (2023).

[3] M. Bleija et al, Nanomaterials 12, 3671 (2022).

Response: We greatly appreciate the insightful and constructive comments and suggestions provided by the reviewers. Based on the reviewer's comments, we have carefully reviewed and organized the corresponding sections of the manuscript. Meanwhile, we also hope that the relevant modifications can meet the requirements of the reviewers. Specifically, as follows:

In Line 59-62 on Page 2

Magnetic metal materials that exhibit significant magnetic loss effects in the low-frequency range, including iron, cobalt, nickel and their alloys or oxides; Including the development and utilization of flexible and biodegradable green polymers (PLA, PBSA, etc.)/carbon-based composites (carbon nanotubes, graphene, etc.) (16-18).

  1. Bertasius, P.; Plyushch, A.; Macutkevic, J.; Banys, J.; Selskis, A.; Platnieks, O.; Gaidukovs, S., Multilayered Composites with Carbon Nanotubes for Electromagnetic Shielding Application. Polymers (Basel) 2023, 15, (4).
  2. Bleija, M.; Platnieks, O.; Macutkevic, J.; Banys, J.; Starkova, O.; Grase, L.; Gaidukovs, S., Poly(Butylene Succinate) Hybrid Multi-Walled Carbon Nanotube/Iron Oxide Nanocomposites: Electromagnetic Shielding and Thermal Properties. Polymers (Basel) 2023, 15, (3).
  3. Bleija, M.; Platnieks, O.; Macutkevic, J.; Starkova, O.; Gaidukovs, S., Comparison of Carbon-Nanoparticle-Filled Poly(Butylene Succinate-co-Adipate) Nanocomposites for Electromagnetic Applications. Nanomaterials (Basel) 2022, 12, (20).

Reviewer 2 Report

Comments and Suggestions for Authors

In this article, the authors have constructed an anisotropic heterojunction interface with magnetic and dielectric coupling based on a honeycomb-like periodic matrix multi-layer array repeating unit. The removal of lignin components from bamboo through oxidation enriches the impregnation pores and uniform adsorption sites of the magnetic medium. This presented research work provides new directions for the diversified development of green biomass and the optimization design of magnetic and dielectric coupling in periodic array structures. The presented research work is very interesting however, I do not recommend its acceptance in the present form. I have following concerns.

 1) The authors must present the research problem clearly and the contribution in the bullet form.

2)  How did the architectural parameters of the simulation model analyse? In particular, I would like to know about its dielectric constant. Was the considered architecture symmetric?

3) The authors provide an equivalent circuit model/parameters of the structure simulated by CST Microwave Suite with simulated parameters.

4)  How does the authors validate results?

Author Response

Dear Editor:

We are grateful for your correspondence and the valuable comments from the reviewer on the quality of our manuscript. We would like to thank the reviewer for providing his concerns, which will further strengthen the presentation of our work. Below we respond to the comments from you and the reviewers of our manuscript in a point-by-point manner.

Reviewer #2:

In this article, the authors have constructed an anisotropic heterojunction interface with magnetic and dielectric coupling based on a honeycomb-like periodic matrix multi-layer array repeating unit. The removal of lignin components from bamboo through oxidation enriches the impregnation pores and uniform adsorption sites of the magnetic medium. This presented research work provides new directions for the diversified development of green biomass and the optimization design of magnetic and dielectric coupling in periodic array structures. The presented research work is very interesting however, I do not recommend its acceptance in the present form. I have following concerns.

Response: We greatly appreciate the insightful and constructive comments and suggestions provided by the reviewers. We believe that the input of the reviewers enabled us to complete a significantly improved manuscript. In response to the comments of the reviewers, we have made extensive revisions to the writing, presentation, and analysis of the research results. These opinions and suggestions not only enable us to provide a highly improved manuscript, but also inspire us to conduct more in-depth research on it in future work. Meanwhile, we also hope that the relevant modifications can meet the requirements of the reviewers.

1) The authors must present the research problem clearly and the contribution in the bullet form.

Response: We greatly appreciate the insightful and constructive comments and suggestions provided by the reviewers. Based on the reviewer's comments, we have carefully reviewed and organized the corresponding sections of the manuscript. Meanwhile, we also hope that the relevant modifications can meet the requirements of the reviewers. Specifically, as follows:

As mentioned by the reviewer, in this manuscript, we removed lignin components that hinder impregnation diffusion inside bamboo through oxidation modification, while selectively adjusting the abundant functional chemical groups on the surface of cellulose and hemicellulose. A series of bamboo-based carbon/Co/CoO multi interface heterojunction electromagnetic absorbers with multi-layer periodic matrix arrays were obtained by introducing cobalt containing solutions of different concentrations into bamboo through vacuum impregnation and in-situ pyrolysis. Develop electromagnetic wave absorbing materials with characteristics such as broadband, lightweight, multiple loss mechanisms, and environmental friendliness. The construction of an anisotropic heterojunction interface with magnetic and dielectric coupling was achieved based on a periodic matrix multi-layer array repeating unit with a honeycomb-like structure. The multi-interface structure of heterojunctions causes multiple scattering and reflection of electromagnetic waves at the interfaces of different phases when propagating inside the multi-level porous structure. The introduction of magnetic nanoparticles forms rich heterojunction interfaces with the carbon matrix, enhancing dipole polarization and interface polarization effects. The electromagnetic wave absorption performance of the material has been significantly improved through interface polarization, enhanced multiple scattering and reflection effects, synergistic effects of magnetic and dielectric properties, and local electric field distortion. By adjusting the appropriate impregnation concentration, the reflection loss RL value of the bamboo-based carbon/Co/CoO composite material obtained reached -54.7 dB (BC-3). In addition, the effective absorption bandwidth of BC-3 reached 5.1 GHz at a matching thickness of 1.55mm. The electromagnetic simulation results of CST far-field radar show that the material exhibits excellent RCS attenuation of 33.3 dB·m2 in the detection angle range of -90º<0<90º. This work provides new directions for the diversified development of green biomass and the optimization design of magnetic and dielectric coupling in periodic array structures.

With the rapid development of modern electrical technology and wireless communication, the density and intensity of electromagnetic radiation are gradually increasing, leading to increasingly serious electromagnetic pollution problems. Long term exposure to high-intensity electromagnetic radiation may pose a threat to human health, electronic device performance, and information security. Therefore, how to effectively absorb and shield electromagnetic waves has become a key issue that urgently needs to be addressed. The existing absorbing materials have many limitations, such as poor low-frequency absorption performance of carbon-based materials and increased electromagnetic wave reflection due to high conductivity. Therefore, it is urgent to develop new electromagnetic wave absorbing materials that are broadband, lightweight, have multiple loss mechanisms, and are environmentally friendly.

In this manuscript, we also listed the shortcomings of current traditional materials and how to overcome them, as follows: conventional carbon-based materials have poor low-frequency absorption performance, a single absorption mechanism is insufficient in broadband absorption performance, and high conductivity leads to increased reflection. The density of foam metal and magnetic metal particles is large and easy to oxidize, the preparation process is complex and energy consumption is large, and the uncontrollable structure reduces the electromagnetic adjustment ability. Conductive polymers are prone to aggregation, and poor dispersibility reduces reflection losses. Considering the efficient multiple loss mechanism of electromagnetic radiation, it is necessary to overcome the shortcomings of traditional absorbing materials in order to further integrate the compositional characteristics of materials and optimize the structure of unit cells. However, there are still significant challenges in constructing lightweight composite absorbers with green, low-cost, and efficient coupling of magnetic and dielectric materials for multi heterojunction interface polarization.

Based on the above analysis, we further introduce the biomass material bamboo. Bamboo has a shorter growth cycle and more abundant resources. Compared to other carbon based materials, bamboo is composed of regularly arranged thin-walled cells and vascular bundles, forming a highly ordered honeycomb like multi-layer matrix array microstructure. This structure endows bamboo-based carbon materials with good mechanical strength and low density. The honeycomb like microstructure itself can provide materials with a natural multi-level pore structure, enriching the multiple reflection and scattering effects of electromagnetic waves. Bamboo has a higher specific surface area and flexibility than other biomass. The abundant natural polymers (cellulose, hemicellulose) inside contain various functional chemical groups (hydroxyl, carboxyl, methoxy, etc.), which provide multiple reaction sites for chemical modification, composite material preparation, and functional applications of bamboo. And after carbonization treatment, the structure can maintain its original strength and toughness well, forming a stable bamboo based carbon material. Therefore, low-cost biomass based lightweight porous electromagnetic absorbers with efficient coupling of magnetic and dielectric media can be obtained through a simple preparation process.

Specific research contributions (key form): (1) Proposed a green and low-cost absorbing material based on bamboo biomass resources: Bamboo has a natural porous honeycomb microstructure, high specific surface area, and flexibility, and after carbonization treatment, it forms a lightweight and strong bamboo-based carbon material. (2) The removal of lignin through oxidation modification has improved the impregnation pores and adsorption performance of bamboo: after removing lignin, more functional chemical groups are exposed in the cellulose and hemicellulose inside the bamboo, providing reaction sites for the introduction of magnetic media and interface polarization in composite materials. (3) By introducing magnetic media such as cobalt and cobalt oxide, efficient coupling between magnetism and dielectric is achieved: cobalt and cobalt oxide are uniformly loaded into bamboo-based carbon materials through vacuum impregnation of cobalt containing solution and in-situ pyrolysis method, forming a multi interface heterojunction structure. (4) By enhancing interface polarization and relaxation through heterojunction interfaces, the electromagnetic wave absorption performance is improved: the introduction of magnetic particles forms an anisotropic heterojunction interface, which enhances charge distribution, dielectric loss, and eddy current effects, thereby improving the absorption efficiency of electromagnetic waves. (5) Optimized the heterojunction structure, achieving a synergistic effect of magnetic loss and dielectric coupling: the optimized structure achieves excellent impedance matching and efficient electromagnetic wave absorption over a wider frequency range, demonstrating good broadband absorption characteristics. (6) This research provides a new direction for the diversified development and periodic array structure design of green biomass materials: it demonstrates a new strategy for electromagnetic wave absorption through biomass based absorbing materials, opening up new ideas for the functional application of green biomass materials.

Correspondingly, in the manuscript, we have made corresponding supplementary modifications as follows:

In Line 76-90 on Page 2

“It is worth noting that magnetic nanoparticles have a higher proportion of surface magnetic atoms due to size and surface effects, resulting in stronger magnetic loss ability. At the same time, the magnetic domain size of nanoparticles is smaller than or close to the single domain size, and under the action of electromagnetic waves, magnetic domain resonance is more significant, which can enhance the absorption of electromagnetic waves. In addition, when lightweight magnetic nanoparticles are combined with other lightweight substrates such as carbon-based materials, the magnetic loss mechanisms such as domain wall movement, hysteresis loss, and eddy current effect in the magnetic material can work together with various loss mechanisms such as dielectric loss and interface polarization to form a composite absorption mechanism. Expand the absorption frequency range of materials. At the same time, it makes the impedance of electromagnetic waves between the material surface and free space more matched, enhancing absorption efficiency. However, there are still significant challenges in constructing lightweight composite absorbers with green, low-cost, and efficient coupling of magnetic and dielectric materials for multi heterojunction interface polarization.”

In Line 114-125 on Page 3

“Uneven distribution or agglomeration of magnetic nanoparticles in porous carbon matrices can lead to weakened interfacial polarization effects, deteriorated impedance matching, reduced multiple scattering and magnetic loss effects, and narrowed effective absorption frequency bands, thereby affecting the material's absorption efficiency and broadband absorption capacity. Reunion will also reduce the lightweight advantage of materials. On the contrary, the introduction of reasonable magnetic particles can generate more anisotropic heterojunction interface structures within bamboo-based carbon. In addition, the introduction of magnetic media can further alleviate the thermal decomposition or expansion of bamboo-based carbon materials at high temperatures, improve the thermal expansion coefficient of the material, and enhance structural stability. Therefore, it is crucial to achieve effective integration of magnetic media inside bamboo.”

2) How did the architectural parameters of the simulation model analyse? In particular, I would like to know about its dielectric constant. Was the considered architecture symmetric?

Response: We greatly appreciate the insightful and constructive comments and suggestions provided by the reviewers. Based on the reviewer's comments, we have carefully reviewed and organized the corresponding sections of the manuscript. Meanwhile, we also hope that the relevant modifications can meet the requirements of the reviewers. Specifically, as follows:

As stated by the reviewer, in this manuscript, in order to further elucidate the electromagnetic wave absorption effect of the BC series samples in practical life. We simulated the loss of electromagnetic waves generated by outdoor far-field radar at different angles using CST (CST Studio Suite) electromagnetic simulation. To ensure the prevention of electromagnetic wave transmission affecting the experimental results, PEC conductor material with dimensions of 100 mm × 100 mm × 2.0 mm (perfect conductor, capable of completely reflecting electromagnetic waves) was added to the back of the sample. Simulate the electromagnetic loss performance of BC series samples by replacing the coating material on the PEC surface. concrete. Set the coating thickness of BC-3 to 2.8mm. The calculation frequency is set to 8.2 GHz. The simulated electromagnetic wave incidence angle range is set to -180° to 180°. At the same time, the Z-axis direction (detection angle of 0°) is defined as the incident direction of electromagnetic waves.

Specifically, in CST electromagnetic simulation, the key parameters of electromagnetic wave absorbing materials are their complex dielectric constant and complex magnetic permeability. These parameters define the electromagnetic properties of the material and affect the propagation and absorption of electromagnetic waves in the material. In CST, it is necessary to define the real and imaginary parts of the dielectric constant of materials to represent their polarization characteristics and electrical energy loss. And the real and imaginary parts of magnetic permeability are used to represent the magnetic response capability and magnetic loss of materials. The definition of conductivity is optional, used to define the electrical conductivity characteristics of materials, especially those that significantly affect losses in metals or conductive materials. In this manuscript, in order to better describe the electromagnetic wave absorption performance of BCs samples, we selected the frequency range of the real and imaginary parts of the dielectric constant and the real and imaginary parts of the magnetic permeability as (8.0-8.55 GHz). By defining this frequency range and setting the specific frequency of 8.2 GHz, we simulated the loss of electromagnetic waves generated by outdoor far-field radar at different angles.

CST simulation software simulation frequency range legend

(2) Geometric structure design: In CST electromagnetic simulation, users are allowed to create complex three-dimensional structures. Especially for designing electromagnetic wave absorbers, the geometric shape and thickness of the material are crucial for the absorption effect. The design of geometric structures usually involves: the thickness of the absorbing layer: the thickness of the material directly affects the absorption behavior of electromagnetic waves. In CST electromagnetic simulation, the wavelength of electromagnetic waves can be matched by optimizing the thickness to achieve maximum absorption. In this manuscript, we define the matching thickness as 2.8 mm based on the electromagnetic wave absorption performance of BC samples at different matching thicknesses, and for better comparison with each other. According to specific circumstances, multi-layer and periodic structures can also be set up. Electromagnetic wave absorbers typically use multiple layers of materials to enhance their absorption performance. In CST electromagnetic simulation, multiple layers of absorbing materials can be designed, and the material parameters and thickness of each layer can be defined separately. In this manuscript, in order to ensure the prevention of electromagnetic wave transmission affecting the experimental results, PEC conductor material with dimensions of 100 mm × 100 mm × 2.0 mm (perfect conductor that can completely reflect electromagnetic waves) was added to the back of the sample. Simulate the electromagnetic loss performance of BC series samples by replacing the coating material on the PEC surface. CST electromagnetic simulation can simulate periodic structures such as metamaterials or photonic crystals. This structure enhances electromagnetic wave absorption through local resonance, which can define the geometric shape, size, and periodicity of periodic elements. In this manuscript, we use Radar Cross Section (RCS) simulation.

(3) Boundary condition setting: The selection of boundary conditions in simulation has a significant impact on the propagation and reflection of electromagnetic waves. In CST electromagnetic simulation, users can choose different boundary conditions. Specifically, it includes open boundaries, periodic boundary conditions, and ideal absorption/reflection boundaries. In this manuscript, in order to ensure the prevention of electromagnetic wave transmission affecting the experimental results, PEC conductor material with dimensions of 100 mm × 100 mm × 2.0 mm (perfect conductor that can completely reflect electromagnetic waves) was added to the back of the sample. PEC conductor material serves as the ideal reflective boundary for the bottom layer. Simulate the electromagnetic loss performance of BC series samples by replacing the coating material on the PEC surface.

(4) Incentive source setting: The excitation method of electromagnetic waves directly affects the simulation results. In CST electromagnetic simulation, different types of electromagnetic wave excitation sources can be selected. Plane Wave Excitation: Used to simulate the incidence of electromagnetic waves in free space. Users can set the polarization direction, incident angle, and frequency range of plane waves. Point or surface source excitation: For the simulation of local electromagnetic fields, point or surface sources can also be used to simulate electromagnetic wave excitation in specific scenarios. In this manuscript, we selected a plane wave excitation source as the excitation source and set the simulated electromagnetic wave incidence angle range to -180 ° to 180 °. At the same time, the Z-axis direction (detection angle of 0 °) is defined as the incident direction of electromagnetic waves. It is possible to systematically study the electromagnetic wave absorption performance of materials at different incident angles. Specifically, we will present the electromagnetic signals on the y-o-z plane in three-dimensional space in detail through two-dimensional curves. As shown in Figure 6e, when the detection angle is within the range of -60 º -60 º, the BC-3 sample has the smallest RCS value. This indicates that almost all electromagnetic energy incident on the material surface in space is effectively absorbed. Figure 6f shows the difference in RCS between the BCs series samples and PEC conductor materials at different detection angles (0° to 80°). It can be seen that the BC-3 sample reaches a maximum of 33.3 dB m2 at a detection angle of 10º. When electromagnetic waves radiate on the surface of materials at an incident angle of 0 degrees, they are more prone to electromagnetic reflection, which is not conducive to absorption. It should be noted that the BC-3 sample still has the maximum RCS attenuation value (24.9 dB m2) at a detection angle of 0 º. It shows excellent electromagnetic wave absorption effect.

(5) Grid division: Grid division is a key step in CST simulation. Due to the use of Finite Integration Technique (FIT) in CST to calculate electromagnetic field distribution, the level of detail in grid partitioning can significantly affect the accuracy and speed of simulation. In electromagnetic wave absorption simulation, it is recommended to use fine grids in the absorption material area to ensure accurate simulation of complex electromagnetic phenomena. In this manuscript, the grid division of the material is shown in the following figure:

CST simulation software grid partitioning parameter legend

Meanwhile, in this manuscript, the model structure considered is symmetrical. Advantages of symmetric models: (1) Simplify simulation calculations: Symmetric structures can utilize symmetry to reduce computational complexity, as only a part of the model can be simulated and then extended to the entire structure through symmetry. This can improve computational efficiency and reduce resource consumption. (2) Uniform scattering characteristics: For certain applications, symmetrical structures provide relatively uniform scattering characteristics, resulting in similar RCS performance in all directions, making them suitable for reducing target detectability. In addition, symmetric structures typically exhibit more stable RCS response when the incident angle of electromagnetic waves changes. This is particularly useful for designing uniform, omnidirectional absorbing materials or stealth systems. (3) Optimize impedance matching and electromagnetic performance. Impedance matching optimization: Symmetrical structures help optimize impedance matching between electromagnetic waves and materials. Better impedance matching can reduce the reflection of electromagnetic waves, increase absorption, and thus improve the stealth capability of materials or structures. Symmetric design makes it easier for the entire structure to achieve impedance matching requirements through uniformly distributed electromagnetic characteristics. Consistent electromagnetic field distribution: A symmetric model can ensure that the electromagnetic field is uniformly distributed within the structure, avoiding the phenomenon of local electric fields being too strong or too weak. This is of great help in improving absorption performance and reducing unnecessary electromagnetic wave reflections.

In summary, in CST Microwave Suite, the architecture parameters of the electromagnetic wave absorption simulation model include the electromagnetic properties of materials, geometric structure design, boundary conditions, excitation sources, grid partitioning, and S-parameter analysis. By accurately analyzing and setting these parameters, users can effectively simulate and optimize the performance of electromagnetic wave absorbers, ultimately achieving wideband and efficient electromagnetic wave absorption effects.

Meanwhile, in this manuscript, it can be seen from Figures 6a-d that the Z-axis direction (detection angle of 0°) is defined as the incident direction of electromagnetic waves. The 3D electromagnetic scattering energy cluster appearing on the surface of the BC-3 sample is the smallest, and the lowest scattering signal indicates that almost all electromagnetic waves are absorbed by the BC-3 material. And the large amount of reflection generated by other samples. Furthermore, we will present the electromagnetic signals on the y-o-z plane in three-dimensional space in detail through two-dimensional curves. As shown in Figure 6e, when the detection angle is within the range of -60º -60º, the BC-3 sample has the smallest RCS value. This indicates that almost all electromagnetic energy incident on the material surface in space is effectively absorbed.

Correspondingly, in the manuscript, we have made corresponding supplementary modifications as follows:

In Line 444-447 on Page 12

“In CST electromagnetic simulation, the key parameters of electromagnetic wave absorbing materials are the complex dielectric constant and complex magnetic permeability of the material. They will affect the propagation and absorption of electromagnetic waves in materials.”

In Line 450-457 on Page 12

“PEC perfect conductor material can serve as the ideal reflection boundary for the bottom layer of the model material, preventing the projection of electromagnetic waves and affecting the simulation experiment results. In addition, the excitation source is set as a plane wave excitation because it can simulate the propagation of electromagnetic waves under far-field conditions, provide uniform illumination, and is suitable for analyzing electromagnetic wave scattering at different angles. The simulation model adopts a symmetrical model, which can better simulate uniform scattering from multiple angles and efficiently present the simulation.”

3) The authors provide an equivalent circuit model/parameters of the structure simulated by CST Microwave Suite with simulated parameters.

Response: We greatly appreciate the insightful and constructive comments and suggestions provided by the reviewers. Based on the reviewer's comments, we have carefully reviewed and organized the corresponding sections of the manuscript. Meanwhile, we also hope that the relevant modifications can meet the requirements of the reviewers. Specifically, as follows:

As stated by the reviewer, in this manuscript, in order to further elucidate the electromagnetic wave absorption effect of the BC series samples in practical life. We simulated the loss of electromagnetic waves generated by outdoor far-field radar at different angles using CST (CST Studio Suite) electromagnetic simulation. In CST electromagnetic simulation, the key parameters of electromagnetic wave absorbing materials are the complex dielectric constant and complex magnetic permeability of the material. They will affect the propagation and absorption of electromagnetic waves in materials. To ensure the prevention of electromagnetic wave transmission affecting the experimental results, PEC conductor material with dimensions of 100 mm × 100 mm × 2.0 mm (perfect conductor, capable of completely reflecting electromagnetic waves) was added to the back of the sample. PEC perfect conductor material can serve as the ideal reflection boundary for the bottom layer of the model material, preventing the projection of electromagnetic waves and affecting the simulation experiment results. In addition, the excitation source is set as a plane wave excitation because it can simulate the propagation of electromagnetic waves under far-field conditions, provide uniform illumination, and is suitable for analyzing electromagnetic wave scattering at different angles. The simulation model adopts a symmetrical model, which can better simulate uniform scattering from multiple angles and efficiently present the simulation. Simulate the electromagnetic loss performance of BC series samples by replacing the coating material on the PEC surface. concrete. Set the coating thickness of BC-3 to 2.8 mm. The calculation frequency is set to 8.2 GHz. The simulated electromagnetic wave incidence angle range is set to -180° to 180°. At the same time, the Z-axis direction (detection angle of 0°) is defined as the incident direction of electromagnetic waves.

In this manuscript, we use a model simulation of RCS (Radar Cross Section), which is an important physical parameter for measuring the reflection ability of objects under electromagnetic wave irradiation. It is particularly used to evaluate the detection ability of radar on target objects. RCS is defined as the ratio of the electromagnetic wave energy reflected by an object in a specific direction to the energy reflected when assuming the target object is a perfect reflector. Simulation can accurately analyze the shape, material properties, and the impact of the angle and frequency of incident electromagnetic waves on the RCS of an object. During the simulation process, numerical methods such as FDTD, FEM, etc. are usually used to simulate the interaction between electromagnetic waves and target objects. Calculate and output the RCS value of the target object within a specific incident angle and frequency range. Generate RCS directional maps to display the scattering characteristics of the target in different directions. RCS simulation is mainly used to evaluate the reflection characteristics of target objects under electromagnetic wave irradiation, helping to optimize the stealth performance or absorption characteristics of objects. Through numerical simulation, designers can simulate the RCS of complex targets, adjust their geometric shape or material properties to reduce their radar detectability.

In the process of radar cross section (RCS) simulation, the S parameter is usually not directly obtained. RCS simulation is mainly used to calculate the scattering characteristics of target objects under electromagnetic wave irradiation, focusing on the object's ability to reflect and scatter electromagnetic waves (i.e. radar cross-sectional area). However, the S-parameter (scattering parameter) is mainly used to describe the reflection and transmission characteristics of electromagnetic waves in transmission systems, such as microwave devices or antenna systems. Therefore, RCS simulation and S-parameter simulation are two types of simulations for different application scenarios.

Therefore, in this manuscript, we mainly calculate and output the RCS values of the target object within a specific incident angle and frequency range through the radar cross section (RCS) simulation process. Generate RCS directional maps to display the scattering characteristics of the target in different directions. From Figure 6a-d, it can be seen that the Z-axis direction (detection angle of 0°) is defined as the incident direction of electromagnetic waves. The 3D electromagnetic scattering energy cluster appearing on the surface of the BC-3 sample is the smallest, and the lowest scattering signal indicates that almost all electromagnetic waves are absorbed by the BC-3 material. And the large amount of reflection generated by other samples. Furthermore, we will present the electromagnetic signals on the y-o-z plane in three-dimensional space in detail through two-dimensional curves. As shown in Figure 6e, when the detection angle is within the range of -60º -60º, the BC-3 sample has the smallest RCS value. This indicates that almost all electromagnetic energy incident on the material surface in space is effectively absorbed. Figure 6f shows the difference in RCS between the BCs series samples and PEC conductor materials at different detection angles (0° to 80°). It can be seen that the BC-3 sample reaches a maximum of 33.3 dB m2 at a detection angle of 10º. When electromagnetic waves radiate on the surface of materials at an incident angle of 0 degrees, they are more prone to electromagnetic reflection, which is not conducive to absorption. It should be noted that the BC-3 sample still has the maximum RCS attenuation value (24.9 dB m2) at a detection angle of 0º. It shows excellent electromagnetic wave absorption effect.

Figure 6. (a-d) Schematic diagram of CST electromagnetic simulation of BC series samples. (e) RCS value curve of BC series samples (detection angle -90º<0<90º). (f) RCS attenuation value image of BC series samples. (g) RCS value curve (detection angle -180º<0<180º).

In future work, we will further study the S-parameter simulation process of CST electromagnetic simulation and analyze and derive the equivalent circuit model of materials during the simulation process.

Therefore, based on the structural characteristics of the material, we provide an equivalent circuit model diagram to describe the electromagnetic wave absorption process of the material through mechanism analysis. Specifically, as shown below:

Figure S8 Material Equivalent Circuit Diagram.

In bamboo-based carbon/Co/CoO multi interface heterojunction electromagnetic absorbing materials, carbonized bamboo and Co/CoO magnetic nanoparticles can be described by an equivalent circuit model. This model is usually composed of resistor (R), capacitor (C), and inductor (L) components, used to represent the electromagnetic properties inside the material, including its absorption, reflection, and loss of electromagnetic waves. The following is a detailed explanation of the equivalent circuit model, as well as the promoting effect of increasing Co/CoO concentration on material properties. Carbonized bamboo (bamboo-based carbon) and Co/CoO magnetic nanoparticles exhibit different electromagnetic properties in an electromagnetic field. The electromagnetic behavior of this composite material can be described by RLC components, as follows: Resistance (R): Used to describe the loss of electromagnetic wave energy inside a material, especially resistive loss. Carbonized bamboo itself has a certain degree of conductivity, and the introduction of Co/CoO nanoparticles further enhances the conductivity. Therefore, the resistance (R) reflects the energy dissipation generated by the current passing through the material. Capacitor (C): Used to describe the polarization ability of materials, that is, the ability of materials to form charge storage under the action of an electric field. The pore structure of bamboo based carbon provides a large specific surface area and can generate strong dipole polarization effects, manifested as the presence of equivalent capacitance. Inductance (L): Co/CoO nanoparticles, as magnetic materials, exhibit magnetic response and eddy current loss effects under the action of electromagnetic waves, which can be described by inductance. As the frequency of electromagnetic waves increases, inductance will limit the variation of current, resulting in the storage and loss of magnetic field energy. In this model, R represents the resistive loss of the composite structure of carbonized bamboo and Co/CoO. C: Representing the dielectric polarization ability of bamboo based carbon. L: Representing the magnetic response and eddy current losses of Co/CoO magnetic nanoparticles. Vin: Input voltage, representing the incident signal of electromagnetic waves. Vout: Output voltage, representing the transmitted or reflected electromagnetic wave signal

As the concentration of Co/CoO magnetic nanoparticles increases, the electromagnetic properties of the material undergo significant changes. These changes are reflected in the R, L, and C components of the equivalent circuit, and their specific promoting effects are as follows: (1) the change in resistance (R) with increasing Co/CoO concentration. At low concentrations: In the case of low Co/CoO particle concentration, the conductivity of bamboo-based carbon dominates, and the conductivity loss in composite materials is relatively small. Therefore, the equivalent resistance is relatively high, and the flow of current inside the material is limited, resulting in less loss of electromagnetic wave energy. At high concentrations: With the increase of Co/CoO concentration, the conductivity of the material is significantly enhanced, especially with the presence of metallic cobalt, which increases the conductivity of the material. The eddy current effect generated by electromagnetic waves in materials is enhanced, resulting in more electromagnetic energy being converted into thermal energy through conductive losses. At this point, the equivalent resistance (R) decreases and the absorption of electromagnetic waves increases.

(2) The variation of capacitance (C) with increasing Co/CoO concentration. At low concentrations: At low concentrations of Co/CoO, the pore structure in bamboo-based carbon materials provides more sites for electric field polarization, resulting in a higher capacitance effect (significant dipole polarization). The dielectric constant of bamboo-based carbon dominates the capacitance characteristics of the material at this stage. At high concentrations: As the concentration of Co/CoO increases, especially when Co/CoO particles aggregate to form a conductive network, the capacitance effect gradually becomes dominated by conductivity. At this point, the interface polarization phenomenon is weakened due to high conductivity, and the polarization ability of the electric dipole decreases, resulting in a decrease in the equivalent capacitance (C). This change is usually more pronounced at high concentrations.

(3) The variation of inductance (L) with increasing Co/CoO concentration. At low concentrations, the magnetic response of Co/CoO magnetic nanoparticles is weak and the inductance effect is not significant. At this point, the magnetic loss in the material is relatively small, and the magnetic permeability (μ) has little effect. At high concentrations: As the concentration of Co/CoO nanoparticles increases, especially when these magnetic particles form a uniform distribution in the material, the magnetic response is significantly enhanced. The eddy current loss, hysteresis loss, and magnetic resonance effect of Co/CoO particles become more pronounced, leading to an increase in equivalent inductance (L). At this point, magnetic losses dominate the material, enhancing its ability to absorb electromagnetic waves.

Correspondingly, in the manuscript, we have made corresponding supplementary modifications as follows:

In Line 480-494 on Page 12

“In bamboo-based carbon/Co/CoO multi-interface heterojunction composite materials, carbonized bamboo and Co/CoO magnetic nanoparticles can be described by an RLC equivalent circuit model. With the increase of Co/CoO concentration from BC-1 to BC-3, conductivity is enhanced, resulting in a decrease in resistance (R) and an increase in electromagnetic wave absorption. The interface polarization decreases, and the capacitance (C) decreases at high concentrations, but the dielectric loss still increases. Magnetic response is enhanced, inductance (L) increases, and magnetic loss dominates electromagnetic wave absorption. The increase in concentration has led to a significant improvement in the electromagnetic wave absorption performance of the material, especially under the synergistic effect of magnetic loss and dielectric loss, which enhances the wideband absorption performance of the material towards electromagnetic waves.”

4)  How does the authors validate results?

Response: We greatly appreciate the insightful and constructive comments and suggestions provided by the reviewers. Based on the reviewer's comments, we have carefully reviewed and organized the corresponding sections of the manuscript. Meanwhile, we also hope that the relevant modifications can meet the requirements of the reviewers. Specifically, as follows:

As stated by the reviewer, in this manuscript, in order to further elucidate the electromagnetic wave absorption effect of the BC series samples in practical life. We simulated the loss of electromagnetic waves generated by outdoor far-field radar at different angles using CST (CST Studio Suite) electromagnetic simulation. After conducting radar cross section (RCS) simulation in CST Microwave Suite, verifying the simulation results is a key step to ensure its accuracy and effectiveness. Verifying RCS simulation results typically requires a combination of different methods, including theoretical calculations, comparison of reference results, and sensitivity analysis of grid partitioning. Specifically, as shown below:

For a flat plate with a length and width much greater than the wavelength, the theoretical calculation formula for RCS is: . Among them, A is the area of the flat plate, and λ is the wavelength of the electromagnetic wave. This formula is applicable when the incident electromagnetic wave is perpendicular to the surface of the flat plate. In CST, the automatic or manual grid partitioning function can be used to adjust the grid density of the target area. Simulate the same structure with different grid densities and observe the changes in RCS results. Low grid density: Simulation results may be inaccurate, especially in targets with complex details such as sharp corners or small structures. High grid density: simulation time and resource consumption increase, but the results tend to stabilize. Frequency range is an important factor in RCS simulation. Ensure that the frequency range used in the simulation is consistent with the frequency of the target application, and verify it by comparing RCS results at different frequencies. Simulate the target in different frequency bands (broadband simulation) and check whether the trend of RCS changing with frequency is reasonable. For example, for typical targets, RCS usually increases with increasing frequency, and peaks may appear near the resonance frequency. RCS is closely related to the incident angle of electromagnetic waves. The RCS distribution may vary at different angles, especially at certain specific angles (such as vertical or horizontal incidence) where significant changes may occur. Set different incident angles in CST, calculate the characteristic curve of RCS changing with incident angle, and analyze whether the results are consistent with expectations. For example, the RCS of a long flat plate under vertical incidence should be greater than that under oblique incidence.

Therefore, in this manuscript, we calculate the characteristic curve of RCS as a function of incident angle by setting different incident angles, and analyze whether the results are consistent with expectations. We will present in detail the electromagnetic signals on the y-o-z plane in three-dimensional space through two-dimensional curves. As shown in Figure 6e, when the detection angle is within the range of -60º -60º, the BC-3 sample has the smallest RCS value. This indicates that almost all electromagnetic energy incident on the material surface in space is effectively absorbed. Figure 6f shows the difference in RCS between the BCs series samples and PEC conductor materials at different detection angles (0° to 80°). It can be seen that the BC-3 sample reaches a maximum of 33.3 dB m2 at a detection angle of 10 º. When electromagnetic waves radiate on the surface of materials at an incident angle of 0 degrees, they are more prone to electromagnetic reflection, which is not conducive to absorption. It should be noted that the BC-3 sample still has the maximum RCS attenuation value (24.9 dB m2) at a detection angle of 0 º. It shows excellent electromagnetic wave absorption effect. By observing the two-dimensional RCS images at Phi of 90º and detection angle range of -180º<0<180º, it can be seen that the BC-3 sample exhibits excellent wide-angle electromagnetic absorption performance. It can absorb electromagnetic radiation energy at most incident angles in space.

Correspondingly, in the manuscript, we have made corresponding supplementary modifications as follows:

In Line 465-470 on Page 12

It is worth noting that RCS is closely related to the incident angle of electromagnetic waves. The RCS distribution may vary at different angles, especially at certain specific angles (such as vertical or horizontal incidence) where significant changes may occur. Set different incident angles in CST, calculate the characteristic curve of RCS changing with incident angle, and analyze whether the results are consistent with expectations.”

Round 2

Reviewer 1 Report

Comments and Suggestions for Authors

I strongly suggest to remove Figs. 5 i, 5j, 5k and 5l and related discussion, because it is not possible to observe several Cole-Cole semicircles in narrow frequency 7-9 GHz. Please fit your obtained results with Cole-Cole equation.

Author Response

Reviewer 1:

I strongly suggest to remove Figs. 5 i, 5j, 5k and 5l and related discussion, because it is not possible to observe several Cole-Cole semicircles in narrow frequency 7-9 GHz. Please fit your obtained results with Cole-Cole equation.

Response: We greatly appreciate the insightful and constructive comments and suggestions provided by the reviewers. We believe that the input of the reviewers enabled us to complete a significantly improved manuscript. In response to the reviewer's comments, we have carefully reviewed and revised the corresponding parts of the article. These opinions and suggestions not only enable us to provide a highly improved manuscript, but also inspire us to conduct more in-depth research on it in future work. Meanwhile, we also hope that the relevant modifications can meet the requirements of the reviewers.

As mentioned by the reviewer, in this manuscript, due to the narrow frequency range of the Cole Cole semicircle in some samples, such as BC-1 appearing in the narrow frequency range of 7-9 GHz. Due to the specific response characteristics of relaxation mechanisms in different frequency ranges, and 7-9 GHz being a relatively narrow frequency range, only one or a few contributions of relaxation mechanisms can usually be seen within this range. Generally speaking, it is difficult to observe multiple Cole-Cole semicircles within a relatively narrow frequency range, possibly due to the frequency response range of the dielectric relaxation process, the distribution of relaxation mechanisms, and the measured frequency bandwidth. In addition, within a narrow frequency band, if there are multiple relaxation mechanisms, their frequency responses may overlap with each other, making it difficult to distinguish the contribution of each mechanism. These relaxation mechanisms will stack together to form a relatively smooth single semicircle or nearly straight line shape. Therefore, the Cole Cole semicircle images presented in these samples cannot be compared and analyzed. For the sake of rigor in the manuscript description, we have removed the relevant legends and descriptions in this section, as shown below:

Figure 5. (a-d)  and  curves. 2D color mapping image of the  value of (e-h) BCs.

At the same time, in order to ensure the rigor of the manuscript description and to make the contextual description of the article smoother, we have added corresponding descriptions. Specifically, as shown below:

In Line 418-430 on Page 11

“Impedance matching characteristics are crucial in electromagnetic wave absorbing materials. Good impedance matching can minimize electromagnetic wave reflection, increase absorption efficiency, and achieve wideband electromagnetic wave absorption. Multiple loss mechanisms (dielectric loss, magnetic loss, interface polarization, eddy current effect, etc.) that enhance the absorption performance simultaneously. At the same time, there are multiple polarization mechanisms within the material (such as dipole polarization and interface polarization), interface effects between magnetic and dielectric materials, multi-scale and multi-layer sub porous structures in bamboo-based carbon/Co/CoO composites, and internal defects in the composite phase structure of the material. These mechanisms have a positive promoting effect on the absorption efficiency of electromagnetic waves in different frequency bands, thereby enabling the material to achieve excellent impedance matching. Therefore, BC-3 achieved satisfactory electromagnetic loss characteristics.”

Reviewer 2 Report

Comments and Suggestions for Authors

In the revised manuscript, the authors have incorporated almost all the comments and suggestion. Therefore, I recommend the acceptance of the manuscript for the publication.

Author Response

Reviewer 2

In the revised manuscript, the authors have incorporated almost all the comments and suggestion. Therefore, I recommend the acceptance of the manuscript for the publication.

Response: Thank you to the reviewer for reviewing and approving the publication of our paper. We greatly appreciate the valuable opinions and suggestions provided by the reviewers during the review process, which have played a crucial role in improving the quality and depth of our work.

Round 3

Reviewer 1 Report

Comments and Suggestions for Authors

Authors make proper corrections according to reviewer remarks and I suggest

to publish the paper as it is.